# CSDI: Conditional Score-based Diffusion Models for Probabilistic Time Series Imputation

**Yusuke Tashiro[123*], Jiaming Song[1], Yang Song[1], Stefano Ermon[1]**
[1]Department of Computer Science, Stanford University, Stanford, CA, USA
[2]Mitsubishi UFJ Trust Investment Technology Institute, Tokyo, Japan
[3]Japan Digital Design, Tokyo, Japan
{ytashiro,tsong,songyang,ermon}@cs.stanford.edu

## Abstract

The imputation of missing values in time series has many applications in healthcare and finance. While autoregressive models are natural candidates for time series imputation, score-based diffusion models have recently outperformed existing counterparts including autoregressive models in many tasks such as image generation and audio synthesis, and would be promising for time series imputation. In this paper, we propose Conditional Score-based Diffusion models for Imputation (CSDI), a novel time series imputation method that utilizes score-based diffusion models conditioned on observed data. Unlike existing score-based approaches, the conditional diffusion model is explicitly trained for imputation and can exploit correlations between observed values. On healthcare and environmental data, CSDI improves by 40-65% over existing probabilistic imputation methods on popular performance metrics. In addition, deterministic imputation by CSDI reduces the error by 5-20% compared to the state-of-the-art deterministic imputation methods. Furthermore, CSDI can also be applied to time series interpolation and probabilistic forecasting, and is competitive with existing baselines. The code is available at https://github.com/ermongroup/CSDI.

## 1 Introduction

Multivariate time series are abundant in real world applications such as finance, meteorology and healthcare. These time series data often contain missing values due to various reasons, including device failures and human errors [1, 2, 3]. Since missing values can hamper the interpretation of a time series, many studies have addressed the task of imputing missing values using machine learning techniques [4, 5, 6]. In the past few years, imputation methods based on deep neural networks have shown great success for both deterministic imputation [7, 8, 9] and probabilistic imputation [10]. These imputation methods typically utilize autoregressive models to deal with time series.

Score-based diffusion models – a class of deep generative models and generate samples by gradually converting noise into a plausible data sample through denoising – have recently achieved state-of-the-art sample quality in many tasks such as image generation [11, 12] and audio synthesis [13, 14], outperforming counterparts including autoregressive models. Diffusion models can also be used to impute missing values by approximating the scores of the posterior distribution obtained from the prior by conditioning on the observed values [12, 15, 16]. While these approximations may work well in practice, they do not correspond to the exact conditional distribution.

In this paper, we propose CSDI, a novel probabilistic imputation method that directly learns the conditional distribution with *conditional* score-based diffusion models. Unlike existing score-based approaches, the conditional diffusion model is designed for imputation and can exploit useful information in observed values. We illustrate the procedure of time series imputation with CSDI in

35th Conference on Neural Information Processing Systems (NeurIPS 2021).

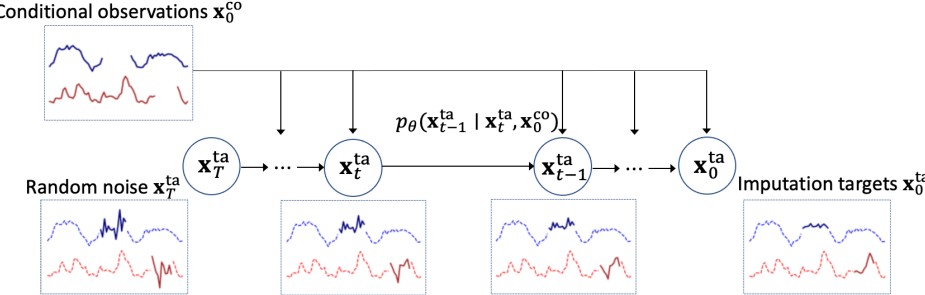

Figure 1: The procedure of time series imputation with CSDI. The reverse process $p_\theta$ gradually converts random noise into plausible time series, conditioned on observed values $\mathbf{x}_0^{co}$. Dashed lines in each box represent observed values, which are plotted in order to show the relationship with generated imputation and not included in each $\mathbf{x}_t^{ta}$.

Figure 1. We start imputation from random noise on the left of the figure and gradually convert the noise into plausible time series through the reverse process $p_\theta$ of the conditional diffusion model. At each step $t$, the reverse process removes noise from the output of the previous step $(t+1)$. Unlike existing score-based diffusion models, the reverse process can take observations (on the top left of the figure) as a conditional input, allowing the model to exploit information in the observations for denoising. We utilize an attention mechanism to capture the temporal and feature dependencies of time series.

For training the conditional diffusion model, we need observed values (i.e., conditional information) and ground-truth missing values (i.e., imputation targets). However, in practice we do not know the ground-truth missing values, or training data may not contain missing values at all. Then, inspired by masked language modeling, we develop a self-supervised training method that separates observed values into conditional information and imputation targets. We note that CSDI is formulated for general imputation tasks, and is not restricted to time series imputation.

Our main contributions are as follows:

- We propose conditional score-based diffusion models for probabilistic imputation (CSDI), and implement CSDI for time series imputation. To train the conditional diffusion model, we develop a self-supervised training method.

- We empirically show that CSDI improves the continuous ranked probability score (CRPS) by 40-65% over existing probabilistic methods on healthcare and environmental data. Moreover, deterministic imputation with CSDI decreases the mean absolute error (MAE) by 5-20% compared to the state-of-the-art methods developed for deterministic imputation.

- We demonstrate that CSDI can also be applied to time series interpolations and probabilistic forecasting, and is competitive with existing baselines designed for these tasks.

## 2   Related works

**Time series imputations with deep learning**   Previous studies have shown deep learning models can capture the temporal dependency of time series and give more accurate imputation than statistical methods. A popular approach using deep learning is to use RNNs, including LSTMs and GRUs, for sequence modeling [17, 8, 7]. Subsequent studies combined RNNs with other methods to improve imputation performance, such as GANs [9, 18, 19] and self-training [20]. Among them, the combination of RNNs with attention mechanisms is particularly successful for imputation and interpolation of time series [21, 22]. While these methods focused on deterministic imputation, GP-VAE [10] has been recently developed as a probabilistic imputation method.

**Score-based generative models**   Score-based generative models, including score matching with Langevin dynamics [23] and denoising diffusion probabilistic models [11], have outperformed existing methods with other deep generative models in many domains, such as images [23, 11],

audio [13, 14], and graphs [24]. Most recently, TimeGrad [25] utilized diffusion probabilistic models for probabilistic time series forecasting. While the method has shown state-of-the-art performance, it cannot be applied to time series imputation due to the use of RNNs to handle past time series.

## 3 Background

### 3.1 Multivariate time series imputation

We consider $N$ *multivariate* time series with missing values. Let us denote the values of each time series as $\mathbf{X} = \{x_{1:K,1:L}\} \in \mathbb{R}^{K \times L}$ where $K$ is the number of features and $L$ is the length of time series. While the length $L$ can be different for each time series, we treat the length of all time series as the same for simplicity, unless otherwise stated. We also denote an observation mask as $\mathbf{M} = \{m_{1:K,1:L}\} \in \{0,1\}^{K \times L}$ where $m_{k,l} = 0$ if $x_{k,l}$ is missing, and $m_{k,l} = 1$ if $x_{k,l}$ is observed. We assume time intervals between two consecutive data entries can be different, and define the timestamps of the time series as $\mathbf{s} = \{s_{1:L}\} \in \mathbb{R}^L$. In summary, each time series is expressed as $\{\mathbf{X}, \mathbf{M}, \mathbf{s}\}$.

Probabilistic time series imputation is the task of estimating the distribution of the missing values of $\mathbf{X}$ by exploiting the observed values of $\mathbf{X}$. We note that this definition of imputation includes other related tasks, such as interpolation, which imputes all features at target time points, and forecasting, which imputes all features at future time points.

### 3.2 Denoising diffusion probabilistic models

Let us consider learning a model distribution $p_\theta(\mathbf{x}_0)$ that approximates a data distribution $q(\mathbf{x}_0)$. Let $\mathbf{x}_t$ for $t = 1, \ldots, T$ be a sequence of latent variables in the same sample space as $\mathbf{x}_0$, which is denoted as $\mathcal{X}$. Diffusion probabilistic models [26] are latent variable models that are composed of two processes: the forward process and the reverse process. The forward process is defined by the following Markov chain:

$$q(\mathbf{x}_{1:T} \mid \mathbf{x}_0) := \prod_{t=1}^{T} q(\mathbf{x}_t \mid \mathbf{x}_{t-1}) \text{ where } q(\mathbf{x}_t \mid \mathbf{x}_{t-1}) := \mathcal{N}\left(\sqrt{1 - \beta_t}\mathbf{x}_{t-1}, \beta_t \mathbf{I}\right) \quad (1)$$

and $\beta_t$ is a small positive constant that represents a noise level. Sampling of $\mathbf{x}_t$ has the closed-form written as $q(\mathbf{x}_t \mid \mathbf{x}_0) = \mathcal{N}(\mathbf{x}_t; \sqrt{\alpha_t}\mathbf{x}_0, (1 - \alpha_t)\mathbf{I})$ where $\hat{\alpha}_t := 1 - \beta_t$ and $\alpha_t := \prod_{i=1}^{t} \hat{\alpha}_i$. Then, $\mathbf{x}_t$ can be expressed as $\mathbf{x}_t = \sqrt{\alpha_t}\mathbf{x}_0 + (1 - \alpha_t)\boldsymbol{\epsilon}$ where $\boldsymbol{\epsilon} \sim \mathcal{N}(\mathbf{0}, \mathbf{I})$. On the other hand, the reverse process denoises $\mathbf{x}_t$ to recover $\mathbf{x}_0$, and is defined by the following Markov chain:

$$p_\theta(\mathbf{x}_{0:T}) := p(\mathbf{x}_T) \prod_{t=1}^{T} p_\theta(\mathbf{x}_{t-1} \mid \mathbf{x}_t), \quad \mathbf{x}_T \sim \mathcal{N}(\mathbf{0}, \mathbf{I}),$$

$$p_\theta(\mathbf{x}_{t-1} \mid \mathbf{x}_t) := \mathcal{N}(\mathbf{x}_{t-1}; \boldsymbol{\mu}_\theta(\mathbf{x}_t, t), \sigma_\theta(\mathbf{x}_t, t)\mathbf{I}). \quad (2)$$

Ho et al. [11] has recently proposed denoising diffusion probabilistic models (DDPM), which considers the following specific parameterization of $p_\theta(\mathbf{x}_{t-1} \mid \mathbf{x}_t)$:

$$\boldsymbol{\mu}_\theta(\mathbf{x}_t, t) = \frac{1}{\alpha_t}\left(\mathbf{x}_t - \frac{\beta_t}{\sqrt{1 - \alpha_t}}\boldsymbol{\epsilon}_\theta(\mathbf{x}_t, t)\right), \; \sigma_\theta(\mathbf{x}_t, t) = \tilde{\beta}_t^{1/2} \text{ where } \tilde{\beta}_t = \begin{cases} \frac{1 - \alpha_{t-1}}{1 - \alpha_t}\beta_t & t > 1 \\ \beta_1 & t = 1 \end{cases}$$

$$(3)$$

where $\boldsymbol{\epsilon}_\theta$ is a trainable denoising function. We denote $\boldsymbol{\mu}_\theta(\mathbf{x}_t, t)$ and $\sigma_\theta(\mathbf{x}_t, t)$ in Eq. (3) as $\boldsymbol{\mu}^{\mathrm{DDPM}}(\mathbf{x}_t, t, \boldsymbol{\epsilon}_\theta(\mathbf{x}_t, t))$ and $\sigma^{\mathrm{DDPM}}(\mathbf{x}_t, t)$, respectively. The denoising function in Eq. (3) also corresponds to a rescaled score model for score-based generative models [23]. Under this parameterization, Ho et al. [11] have shown that the reverse process can be trained by solving the following optimization problem:

$$\min_\theta \mathcal{L}(\theta) := \min_\theta \mathbb{E}_{\mathbf{x}_0 \sim q(\mathbf{x}_0), \boldsymbol{\epsilon} \sim \mathcal{N}(\mathbf{0}, \mathbf{I}), t}||\boldsymbol{\epsilon} - \boldsymbol{\epsilon}_\theta(\mathbf{x}_t, t)||_2^2 \quad \text{where } \mathbf{x}_t = \sqrt{\alpha_t}\mathbf{x}_0 + (1 - \alpha_t)\boldsymbol{\epsilon}. \quad (4)$$

The denoising function $\boldsymbol{\epsilon}_\theta$ estimates the noise vector $\boldsymbol{\epsilon}$ that was added to its noisy input $\mathbf{x}_t$. This training objective also be viewed as a weighted combination of denoising score matching used for training score-based generative models [23, 27, 12]. Once trained, we can sample $\mathbf{x}_0$ from Eq. (2). We provide the details of DDPM in Appendix A.

### 3.3 Imputation with diffusion models

Here, we focus on general imputation tasks that are not restricted to time series imputation. Let us consider the following imputation problem: given a sample $\mathbf{x}_0$ which contains missing values, we generate imputation targets $\mathbf{x}_0^{\mathrm{ta}} \in \mathcal{X}^{\mathrm{ta}}$ by exploiting conditional observations $\mathbf{x}_0^{\mathrm{co}} \in \mathcal{X}^{\mathrm{co}}$, where $\mathcal{X}^{\mathrm{ta}}$ and $\mathcal{X}^{\mathrm{co}}$ are a part of the sample space $\mathcal{X}$ and vary per sample. Then, the goal of probabilistic imputation is to estimate the true conditional data distribution $q(\mathbf{x}_0^{\mathrm{ta}} \mid \mathbf{x}_0^{\mathrm{co}})$ with a model distribution $p_\theta(\mathbf{x}_0^{\mathrm{ta}} \mid \mathbf{x}_0^{\mathrm{co}})$. We typically impute all missing values using all observed values, and set all observed values as $\mathbf{x}_0^{\mathrm{co}}$ and all missing values as $\mathbf{x}_0^{\mathrm{ta}}$, respectively. Note that time series imputation in Section 3.1 can be considered as a special case of this task.

Let us consider modeling $p_\theta(\mathbf{x}_0^{\mathrm{ta}} \mid \mathbf{x}_0^{\mathrm{co}})$ with a diffusion model. In the unconditional case, the reverse process $p_\theta(\mathbf{x}_{0:T})$ is used to define the final data model $p_\theta(\mathbf{x}_0)$. Then, a natural approach is to extend the reverse process in Eq. (2) to a conditional one:

$$p_\theta(\mathbf{x}_{0:T}^{\mathrm{ta}} \mid \mathbf{x}_0^{\mathrm{co}}) := p(\mathbf{x}_T^{\mathrm{ta}}) \prod_{t=1}^{T} p_\theta(\mathbf{x}_{t-1}^{\mathrm{ta}} \mid \mathbf{x}_t^{\mathrm{ta}}, \mathbf{x}_0^{\mathrm{co}}), \quad \mathbf{x}_T^{\mathrm{ta}} \sim \mathcal{N}(\mathbf{0}, \mathbf{I}),$$

$$p_\theta(\mathbf{x}_{t-1}^{\mathrm{ta}} \mid \mathbf{x}_t^{\mathrm{ta}}, \mathbf{x}_0^{\mathrm{co}}) := \mathcal{N}(\mathbf{x}_{t-1}^{\mathrm{ta}}; \boldsymbol{\mu}_\theta(\mathbf{x}_t^{\mathrm{ta}}, t \mid \mathbf{x}_0^{\mathrm{co}}), \sigma_\theta(\mathbf{x}_t^{\mathrm{ta}}, t \mid \mathbf{x}_0^{\mathrm{co}})\mathbf{I}). \tag{5}$$

However, existing diffusion models are generally designed for data generation and do not take conditional observations $\mathbf{x}_0^{\mathrm{co}}$ as inputs. To utilize diffusion models for imputation, previous studies [12, 15, 16] approximated the conditional reverse process $p_\theta(\mathbf{x}_{t-1}^{\mathrm{ta}} \mid \mathbf{x}_t^{\mathrm{ta}}, \mathbf{x}_0^{\mathrm{co}})$ with the reverse process in Eq. (2). With this approximation, in the reverse process they add noise to both the target and the conditional observations $\mathbf{x}_0^{\mathrm{co}}$. While this approach can impute missing values, the added noise can harm useful information in the observations. This suggests that modeling $p_\theta(\mathbf{x}_{t-1}^{\mathrm{ta}} \mid \mathbf{x}_t^{\mathrm{ta}}, \mathbf{x}_0^{\mathrm{co}})$ without approximations can improve the imputation quality. Hereafter, we call the model defined in Section 3.2 as the unconditional diffusion model.

## 4 Conditional score-based diffusion model for imputation (CSDI)

In this section, we propose CSDI, a novel imputation method based on a conditional score-based diffusion model. The conditional diffusion model allows us to exploit useful information in observed values for accurate imputation. We provide the reverse process of the conditional diffusion model, and then develop a self-supervised training method. We note that CSDI is not restricted to time series.

### 4.1 Imputation with CSDI

We focus on the conditional diffusion model with the reverse process in Eq. (5) and aim to model the conditional distribution $p(\mathbf{x}_{t-1}^{\mathrm{ta}} \mid \mathbf{x}_t^{\mathrm{ta}}, \mathbf{x}_0^{\mathrm{co}})$ without approximations. Specifically, we extend the parameterization of DDPM in Eq. (3) to the conditional case. We define a conditional denoising function $\boldsymbol{\epsilon}_\theta : (\mathcal{X}^{\mathrm{ta}} \times \mathbb{R} \mid \mathcal{X}^{\mathrm{co}}) \rightarrow \mathcal{X}^{\mathrm{ta}}$, which takes conditional observations $\mathbf{x}_0^{\mathrm{co}}$ as inputs. Then, we consider the following parameterization with $\boldsymbol{\epsilon}_\theta$:

$$\boldsymbol{\mu}_\theta(\mathbf{x}_t^{\mathrm{ta}}, t \mid \mathbf{x}_0^{\mathrm{co}}) = \boldsymbol{\mu}^{\mathrm{DDPM}}(\mathbf{x}_t^{\mathrm{ta}}, t, \boldsymbol{\epsilon}_\theta(\mathbf{x}_t^{\mathrm{ta}}, t \mid \mathbf{x}_0^{\mathrm{co}})), \quad \sigma_\theta(\mathbf{x}_t^{\mathrm{ta}}, t \mid \mathbf{x}_0^{\mathrm{co}}) = \sigma^{\mathrm{DDPM}}(\mathbf{x}_t^{\mathrm{ta}}, t) \tag{6}$$

where $\boldsymbol{\mu}^{\mathrm{DDPM}}$ and $\sigma^{\mathrm{DDPM}}$ are the functions defined in Section 3.2. Given the function $\boldsymbol{\epsilon}_\theta$ and data $\mathbf{x}_0$, we can sample $\mathbf{x}_0^{\mathrm{ta}}$ using the reverse process in Eq. (5) and Eq. (6). For the sampling, we set all observed values of $\mathbf{x}_0$ as conditional observations $\mathbf{x}_0^{\mathrm{co}}$ and all missing values as imputation targets $\mathbf{x}_0^{\mathrm{ta}}$. Note that the conditional model is reduced to the unconditional one under no conditional observations and can also be used for data generation.

### 4.2 Training of CSDI

Since Eq. (6) uses the same parameterization as Eq. (3) and the difference between Eq. (3) and Eq. (6) is only the form of $\boldsymbol{\epsilon}_\theta$, we can follow the training procedure for the unconditional model in Section 3.2. Namely, given conditional observations $\mathbf{x}_0^{\mathrm{co}}$ and imputation targets $\mathbf{x}_0^{\mathrm{ta}}$, we sample noisy targets $\mathbf{x}_t^{\mathrm{ta}} = \sqrt{\alpha_t}\mathbf{x}_0^{\mathrm{ta}} + (1 - \alpha_t)\boldsymbol{\epsilon}$, and train $\boldsymbol{\epsilon}_\theta$ by minimizing the following loss function:

$$\min_\theta \mathcal{L}(\theta) := \min_\theta \mathbb{E}_{\mathbf{x}_0 \sim q(\mathbf{x}_0), \boldsymbol{\epsilon} \sim \mathcal{N}(\mathbf{0}, \mathbf{I}), t} ||(\boldsymbol{\epsilon} - \boldsymbol{\epsilon}_\theta(\mathbf{x}_t^{\mathrm{ta}}, t \mid \mathbf{x}_0^{\mathrm{co}}))||_2^2 \tag{7}$$

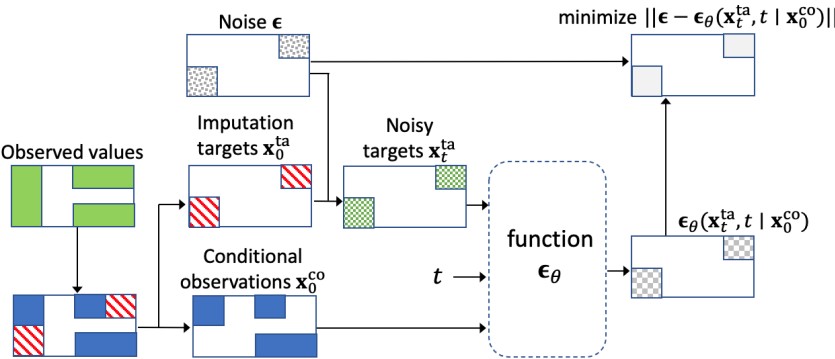

Figure 2: The self-supervised training procedure of CSDI. On the middle left rectangle, the green and white areas represent observed and missing values, respectively. The observed values are separated into red imputation targets $\mathbf{x}_0^{\text{ta}}$ and blue conditional observations $\mathbf{x}_0^{\text{co}}$, and used for training of $\epsilon_\theta$. The colored areas in each rectangle mean the existence of values.

Table 1: Imputation targets $\mathbf{x}_0^{\text{ta}}$ and conditional observations $\mathbf{x}_0^{\text{co}}$ for CSDI at training and sampling.

|  | imputation targets $\mathbf{x}_0^{\text{ta}}$ | conditional observations $\mathbf{x}_0^{\text{co}}$ |
| --- | --- | --- |
| sampling (imputation) | all missing values | all observed values |
| training | a subset of the observed values (sampled by a target choice strategy) | the remaining observed values |

where the dimension of $\epsilon$ corresponds to that of the imputation targets $\mathbf{x}_0^{\text{ta}}$.

However, this training procedure has an issue. Since we do not know the ground-truth missing values in practice, it is not clear how to select $\mathbf{x}_0^{\text{co}}$ and $\mathbf{x}_0^{\text{ta}}$ from a training sample $\mathbf{x}_0$. To address this issue, we develop a self-supervised learning method inspired by masked language modeling [28]. We illustrate the training procedure in Figure 2. Given a sample $\mathbf{x}_0$, we separate observed values of $\mathbf{x}_0$ into two parts, and set one of them as imputation targets $\mathbf{x}_0^{\text{ta}}$ and the other as conditional observations $\mathbf{x}_0^{\text{co}}$. We choose the targets $\mathbf{x}_0^{\text{ta}}$ through a target choice strategy, which is discussed in Section 4.3. Then, we sample noisy targets $\mathbf{x}_t^{\text{ta}}$ and train $\epsilon_\theta$ by solving Eq. (7). We summarize how we set $\mathbf{x}_0^{\text{co}}$ and $\mathbf{x}_0^{\text{ta}}$ for training and sampling in Table 1. We also provide the algorithm of training and sampling in Appendix B.1.

### 4.3 Choice of imputation targets in self-supervised learning

In the proposed self-supervised learning, the choice of imputation targets is important. We provide four target choice strategies depending on what is known about the missing patterns in the test dataset. We describe the algorithm for these strategies in Appendix B.2.

(1) *Random* strategy : this strategy is used when we do not know about missing patterns, and randomly chooses a certain percentage of observed values as imputation targets. The percentage is sampled from $[0\%, 100\%]$ to adapt to various missing ratios in the test dataset.

(2) *Historical* strategy: this strategy exploits missing patterns in the training dataset. Given a training sample $\mathbf{x}_0$, we randomly draw another sample $\tilde{\mathbf{x}}_0$ from the training dataset. Then, we set the intersection of the observed indices of $\mathbf{x}_0$ and the missing indices of $\tilde{\mathbf{x}}_0$ as imputation targets. The motivation of this strategy comes from structured missing patterns in the real world. For example, missing values often appear consecutively in time series data. When missing patterns in the training and test dataset are highly correlated, this strategy helps the model learn a good conditional distribution.

(3) *Mix* strategy: this strategy is the mix of the above two strategies. The historical strategy may lead to overfitting to missing patterns in the training dataset. The *Mix* strategy can benefit from generalization by the random strategy and structured missing patterns by the historical strategy.

(4) *Test* pattern strategy: when we know the missing patterns in the test dataset, we just set the patterns as imputation targets. For example, this strategy is used for time series forecasting, since the missing patterns in the test dataset are fixed to given future time points.

## 5  Implementation of CSDI for time series imputation

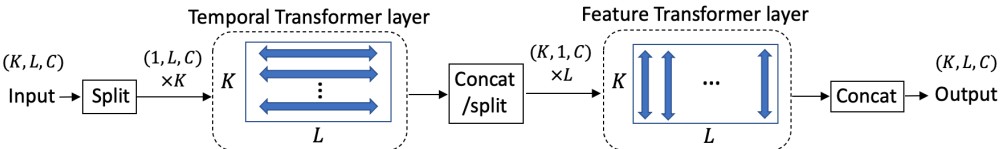

Figure 3: The architecture of 2D attention. Given a tensor with $K$ features, $L$ length, and $C$ channels, the temporal Transformer layer takes tensors with $(1, L, C)$ shape as inputs and learns temporal dependency. The feature Transformer layer takes tensors with $(K, 1, C)$ shape as inputs and learns feature dependency. The output shape of each layer is the same as the input shape.

In this section, we implement CSDI for time series imputation. For the implementation, we need the inputs and the architecture of $\epsilon_\theta$.

First, we describe how we process time series data as inputs for CSDI. As defined in Section 3.1, a time series is denoted as $\{\mathbf{X}, \mathbf{M}, \mathbf{s}\}$, and the sample space $\mathcal{X}$ of $\mathbf{X}$ is $\mathbb{R}^{K \times L}$. We want to handle $\mathbf{X}$ in the sample space $\mathbb{R}^{K \times L}$ for learning dependencies in a time series using a neural network, but the conditional denoising function $\epsilon_\theta$ takes inputs $\mathbf{x}_t^{\mathrm{ta}}$ and $\mathbf{x}_0^{\mathrm{co}}$ in varying sample spaces that are a part of $\mathcal{X}$ as shown in white areas of $\mathbf{x}_t^{\mathrm{ta}}$ and $\mathbf{x}_0^{\mathrm{co}}$ in Figure 2. To address this issue, we adjust the conditional denoising function $\epsilon_\theta$ to inputs in the fixed sample space $\mathbb{R}^{K \times L}$. Concretely, we fix the shape of the inputs $\mathbf{x}_t^{\mathrm{ta}}$ and $\mathbf{x}_0^{\mathrm{co}}$ to $(K \times L)$ by applying zero padding to $\mathbf{x}_t^{\mathrm{ta}}$ and $\mathbf{x}_0^{\mathrm{co}}$. In other words, we set zero values to white areas for $\mathbf{x}_t^{\mathrm{ta}}$ and $\mathbf{x}_0^{\mathrm{co}}$ in Figure 2. To indicate which indices are padded, we introduce the conditional mask $\mathbf{m}^{\mathrm{co}} \in \{0, 1\}^{K \times L}$ as an additional input to $\epsilon_\theta$, which corresponds to $\mathbf{x}_0^{\mathrm{co}}$ and takes value 1 for indices of conditional observations. For ease of handling, we also fix the output shape in the sample space $\mathbb{R}^{K \times L}$ by applying zero padding. Then, the conditional denoising function $\epsilon_\theta(\mathbf{x}_t^{\mathrm{ta}}, t \mid \mathbf{x}_0^{\mathrm{co}}, \mathbf{m}^{\mathrm{co}})$ can be written as $\epsilon_\theta : (\mathbb{R}^{K \times L} \times \mathbb{R} \mid \mathbb{R}^{K \times L} \times \{0, 1\}^{K \times L}) \to \mathbb{R}^{K \times L}$. We discuss the effect of this adjustment on training and sampling in Appendix D.

Under the adjustment, we set conditional observations $\mathbf{x}_0^{\mathrm{co}}$ and imputation targets $\mathbf{x}_0^{\mathrm{ta}}$ for time series imputation by following Table 1. At sampling time, since conditional observations $\mathbf{x}_0^{\mathrm{co}}$ are all observed values, we set $\mathbf{m}^{\mathrm{co}} = \mathbf{M}$ and $\mathbf{x}_0^{\mathrm{co}} = \mathbf{m}^{\mathrm{co}} \odot \mathbf{X}$ where $\odot$ represents element-wise products. For training, we sample $\mathbf{x}_0^{\mathrm{ta}}$ and $\mathbf{x}_0^{\mathrm{co}}$ through a target choice strategy, and set the indices of $\mathbf{x}_0^{\mathrm{co}}$ as $\mathbf{m}^{\mathrm{co}}$. Then, $\mathbf{x}_0^{\mathrm{co}}$ is written as $\mathbf{x}_0^{\mathrm{co}} = \mathbf{m}^{\mathrm{co}} \odot \mathbf{X}$ and $\mathbf{x}_0^{\mathrm{ta}}$ is obtained as $\mathbf{x}_0^{\mathrm{ta}} = (\mathbf{M} - \mathbf{m}^{\mathrm{co}}) \odot \mathbf{X}$.

Next, we describe the architecture of $\epsilon_\theta$. We adopt the architecture in DiffWave [13] as the base, which is composed of multiple residual layers with residual channel $C$. We refine this architecture for time series imputation. We set the diffusion step $T = 50$. We discuss the main differences from DiffWave (see Appendix E.1 for the whole architecture and details).

**Attention mechanism**   To capture temporal and feature dependencies of multivariate time series, we utilize a two dimensional attention mechanism in each residual layer instead of a convolution architecture. As shown in Figure 3, we introduce temporal Transformer layer and a feature Transformer layer, which are 1-layer Transformer encoders. The temporal Transformer layer takes tensors for each feature as inputs to learn temporal dependency, whereas the feature Transformer layer takes tensors for each time point as inputs to learn temporal dependency.

Note that while the length $L$ can be different for each time series as mentioned in Section 3.1, the attention mechanism allows the model to handle various lengths. For batch training, we apply zero padding to each sequence so that the lengths of the sequences are the same.

**Side information**   In addition to the arguments of $\epsilon_\theta$, we provide some side information as additional inputs to the model. First, we use time embedding of $\mathbf{s} = \{s_{1:L}\}$ to learn the temporal dependency. Following previous studies [29, 30], we use 128-dimensions temporal embedding. Second, we exploit categorical feature embedding for $K$ features, where the dimension is 16.

## 6   Experimental results

In this section, we demonstrate the effectiveness of CSDI for time series imputation. Since CSDI can be applied to other related tasks such as interpolation and forecasting, we also evaluate CSDI for these tasks to show the flexibility of CSDI. Due to the page limitation, we provide the detailed setup for experiments including train/validation/test splits and hyperparameters in Appendix E.2.

### 6.1   Time series imputation

**Dataset and experiment settings**   We run experiments for two datasets. The first one is the healthcare dataset in PhysioNet Challenge 2012 [1], which consists of 4000 clinical time series with 35 variables for 48 hours from intensive care unit (ICU). Following previous studies [7, 8], we process the dataset to hourly time series with 48 time steps. The processed dataset contains around 80% missing values. Since the dataset has no ground-truth, we randomly choose 10/50/90% of observed values as ground-truth on the test data.

The second one is the air quality dataset [2]. Following previous studies [7, 21], we use hourly sampled PM2.5 measurements from 36 stations in Beijing for 12 months and set 36 consecutive time steps as one time series. There are around 13% missing values and the missing patterns are not random. The dataset contains artificial ground-truth, whose missing patterns are also structured.

For both dataset, we run each experiment five times. As the target choice strategy for training, we adopt the random strategy for the healthcare dataset and the mix of the random and historical strategy for the air quality dataset, based on the missing patterns of each dataset.

**Results of probabilistic imputation**   CSDI is compared with three baselines. 1) Multitask GP [31]: the method learns the covariance between timepoints and features simultaneously. 2) GP-VAE [10]: the method showed the state-of-the-art results for probabilistic imputation. 3) V-RIN [32]: a deterministic imputation method that uses the uncertainty quantified by VAE to improve imputation. For V-RIN, we regard the quantified uncertainty as probabilistic imputation. In addition, we compare CSDI with imputation using the unconditional diffusion model in order to show the effectiveness of the conditional one (see Appendix C for training and imputation with the unconditional diffusion model).

We first show quantitative results. We adopt the continuous ranked probability score (CRPS) [33] as the metric, which is freuquently used for evaluating probabilistic time series forecasting and measures the compatibility of an estimated probability distribution with an observation. We generate 100 samples to approximate the probability distribution over missing values and report the normalized average of CRPS for all missing values following previous studies [34] (see Appendix E.3 for details of the computation).

Table 2: Comparing CRPS for probabilistic imputation baselines and CSDI (lower is better). We report the mean and the standard error of CRPS for five trials.

|  | healthcare | | | air quality |
|---|---|---|---|---|
|  | 10% missing | 50% missing | 90% missing |  |
| Multitask GP [31] | 0.489(0.005) | 0.581(0.003) | 0.942(0.010) | 0.301(0.003) |
| GP-VAE [10] | 0.574(0.003) | 0.774(0.004) | 0.998(0.001) | 0.397(0.009) |
| V-RIN [32] | 0.808(0.008) | 0.831(0.005) | 0.922(0.003) | 0.526(0.025) |
| unconditional | 0.360(0.007) | 0.458(0.008) | 0.671(0.007) | 0.135(0.001) |
| **CSDI** (proposed) | **0.238(0.001)** | **0.330(0.002)** | **0.522(0.002)** | **0.108(0.001)** |

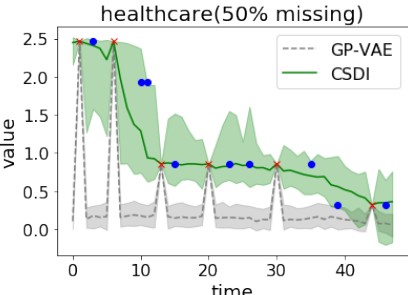
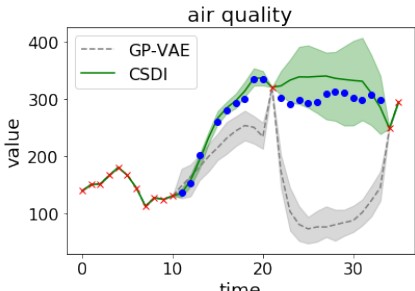

Figure 4: Examples of probabilistic time series imputation for the healthcare dataset with 50% missing (left) and the air quality dataset (right). The red crosses show the observed values and the blue circles show the ground-truth imputation targets. For each method, median values of imputations are shown as the line and 5% and 95% quantiles are shown as the shade.

Table 2 represents CRPS for each method. CSDI reduces CRPS by 40-65% compared to the existing baselines for both datasets. This indicates that CSDI generates more realistic distributions than other methods. We also observe that the imputation with CSDI outperforms that with the unconditional model. This suggests CSDI benefits from explicitly modeling the conditional distribution.

We provide imputation examples in Figure 4. For the air quality dataset, CSDI (green solid line) provides accurate imputations with high confidence, while those by GP-VAE (gray dashed line) are far from ground-truth. CSDI also gives reasonable imputations for the healthcare dataset. These results indicate that CSDI exploits temporal and feature dependencies to provide accurate imputations. We give more examples in Appendix G.

Table 3: Comparing MAE for deterministic imputation methods and CSDI. We report the mean and the standard error for five trials. The asterisks mean the results of the method are cited from the original paper.

|  | healthcare | | | air quality |
|---|---|---|---|---|
|  | 10% missing | 50% missing | 90% missing |  |
| V-RIN [32] | 0.271(0.001) | 0.365(0.002) | 0.606(0.006) | 25.4(0.62) |
| BRITS [7] | 0.284(0.001) | 0.368(0.002) | 0.517(0.002) | 14.11(0.26) |
| BRITS [7] (*) | 0.278 | – | – | 11.56 |
| GLIMA [21] (*) | 0.265 | – | – | 10.54 |
| RDIS [20] | 0.319(0.002) | 0.419(0.002) | 0.631(0.002) | 22.11(0.35) |
| unconditional | 0.326(0.008) | 0.417(0.010) | 0.625(0.010) | 12.13(0.07) |
| **CSDI** (proposed) | **0.217(0.001)** | **0.301(0.002)** | **0.481(0.003)** | **9.60(0.04)** |

**Results of deterministic imputation** We demonstrate that CSDI also provides accurate deterministic imputations, which are obtained as the median of 100 generated samples. We compare CSDI with four baselines developed for deterministic imputation including GLIMA [21], which combined recurrent imputations with an attention mechanism to capture temporal and feature dependencies and showed the state-of-the-art performance. These methods are based on autoregressive models. We use the original implementations except RDIS.

We evaluate each method by the mean absolute error (MAE). In Table 3, CSDI improves MAE by 5-20% compared to the baselines. This suggests that the conditional diffusion model is effective to learn temporal and feature dependencies for imputation. For the healthcare dataset, the gap between the baselines and CSDI is particularly significant when the missing ratio is small, because more observed values help CSDI capture dependencies.

Table 4: Comparing the state-of-the-art interpolation methods with CSDI for the healthcare dataset. We report the mean and the standard error of CRPS for five trials.

|  | 10% missing | 50% missing | 90% missing |
|---|---|---|---|
| Latent ODE [35] | 0.700(0.002) | 0.676(0.003) | 0.761(0.010) |
| mTANs [22] | 0.526(0.004) | 0.567(0.003) | 0.689(0.015) |
| **CSDI** (proposed) | **0.380(0.002)** | **0.418(0.001)** | **0.556(0.003)** |

## 6.2 Interpolation of irregularly sampled time series

**Dataset and experiment settings**   We use the same healthcare dataset as the previous section, but process the dataset as irregularly sampled time series, following previous studies [22, 35]. Since the dataset has no ground-truth, we randomly choose 10/50/90% of *time* points and use observed values at these time points as ground-truth on the test data. As the target choice strategy for training, we adopt the random strategy, which is adjusted for interpolation so that some *time* points are sampled.

**Results**   We compare CSDI with two baselines including mTANs [22], which utilized an attention mechanism and showed state-of-the-art results for the interpolation of irregularly sampled time series. We generate 100 samples to approximate the probability distribution as with the previous section. The result is shown in Table 4. CSDI outperforms the baselines for all cases.

Table 5: Comparing probabilistic forecasting methods with CSDI. We report the mean and the standard error of CRPS-sum for three trials. The baseline results are cited from the original paper. 'TransMAF' is the abbreviation for 'Transformer MAF'.

|  | solar | electricity | traffic | taxi | wiki |
|---|---|---|---|---|---|
| GP-copula [34] | 0.337(0.024) | 0.024(0.002) | 0.078(0.002) | 0.208(0.183) | 0.086(0.004) |
| TransMAF [36] | 0.301(0.014) | 0.021(0.000) | 0.056(0.001) | 0.179(0.002) | 0.063(0.003) |
| TLAE [37] | **0.124(0.033)** | 0.040(0.002) | 0.069(0.001) | 0.130(0.006) | 0.241(0.001) |
| TimeGrad [25] | 0.287(0.020) | 0.021(0.001) | 0.044(0.006) | **0.114(0.020)** | 0.049(0.002) |
| **CSDI** (proposed) | 0.298(0.004) | **0.017(0.000)** | **0.020(0.001)** | 0.123(0.003) | **0.047(0.003)** |

## 6.3 Time series Forecasting

**Dataset and Experiment settings**   We use five datasets that are commonly used for evaluating probabilistic time series forecasting. Each dataset is composed of around 100 to 2000 features. We predict all features at future time steps using past time series. We use the same prediction steps as previous studies [34, 37]. For the target choice strategy, we adopt the *Test* pattern strategy.

**Results**   We compare CSDI with four baselines. Specifically, TimeGrad [25] combined the diffusion model with a RNN-based encoder. We evaluate each method for CRPS-sum, which is CRPS for the distribution of the sum of all time series across $K$ features and accounts for joint effect (see Appendix E.3 for details).

In Table 5, CSDI outperforms the baselines for electricity and traffic datasets, and is competitive with the baselines as a whole. The advantage of CSDI over baselines for forecasting is smaller than that for imputation in Section 6.1. We hypothesize it is because the datasets for forecasting seldom contains missing values and are suitable for existing encoders including RNNs. For imputation, it is relatively difficult for RNNs to handle time series due to missing values.

## 7 Conclusion

In this paper, we have proposed CSDI, a novel approach to impute multivariate time series with conditional diffusion models. We have shown that CSDI outperforms the existing probabilistic and deterministic imputation methods.

There are some interesting directions for future work. One direction is to improve the computation efficiency. While diffusion models generate plausible samples, sampling is generally slower than other generative models. To mitigate the issue, several recent studies leverage an ODE solver to accelerate the sampling procedure [12, 38, 13]. Combining our method with these approaches would likely improve the sampling efficiency.

Another direction is to extend CSDI to downstream tasks such as classifications. Many previous studies have shown that accurate imputation improves the performance on downstream tasks [7, 18, 22]. Since conditional diffusion models can learn temporal and feature dependencies with uncertainty, joint training of imputations and downstream tasks using conditional diffusion models would be helpful to improve the performance of the downstream tasks.

Finally, although our focus was on time series, it would be interesting to explore CSDI as imputation technique on other modalities.

## Acknowledgements and Disclosure of Funding

This research was supported by NSF(#1651565, #1522054, #1733686), ONR (N000141912145), AFOSR (FA95501910024), ARO (W911NF-21-1-0125) and Sloan Fellowship.

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
