# A  Details of denoising diffusion probabilistic models

In this section, we describe the details of denoising diffusion probabilistic models in Section 3.2.

Diffusion probabilistic models [26] are latent variable models that are composed of two processes: the forward process and the reverse process. The forward process and the reverse process are defined by Eq. (1) and 2, respectively. Then, the parameters $\theta$ are learned by maximizing variational lower bound (ELBO) of likelihood $p_\theta(\mathbf{x}_{0:T})$:

$$\mathbb{E}_{q(\mathbf{x}_0)}[\log p_\theta(\mathbf{x}_0)] \geq \mathbb{E}_{q(\mathbf{x}_0, \mathbf{x}_1, \ldots, \mathbf{x}_T)}[\log p_\theta(\mathbf{x}_{0:T}) - \log q(\mathbf{x}_{1:T} \mid \mathbf{x}_0)] := \text{ELBO}. \tag{8}$$

To analyse this ELBO, Ho et al. [11] proposed denoising diffusion probabilistic models (DDPM), which considered the parameterization given by Eq. (3). Under the parameterization, Ho et al. [11] showed ELBO satisfies the following equation:

$$-\text{ELBO} = c + \sum_{t=1}^{T} \kappa_t \mathbb{E}_{\mathbf{x}_0 \sim q(\mathbf{x}_0), \boldsymbol{\epsilon} \sim \mathcal{N}(\mathbf{0}, \mathbf{I})} ||\boldsymbol{\epsilon} - \boldsymbol{\epsilon}_\theta(\sqrt{\alpha_t}\mathbf{x}_0 + (1 - \alpha_t)\boldsymbol{\epsilon}, t)||_2^2 \tag{9}$$

where $c$ is a constant and $\{\kappa_{1:T}\}$ are positive coefficients depending on $\alpha_{1:T}$ and $\beta_{1:T}$. The diffusion process can be trained by minimizing Eq. (9). In addition, Ho et al. [11] found that minimizing the following unweighted version of ELBO leads to good sample quality:

$$\min_\theta L(\theta) := \min_\theta \mathbb{E}_{\mathbf{x}_0 \sim q(\mathbf{x}_0), \boldsymbol{\epsilon} \sim \mathcal{N}(\mathbf{0}, \mathbf{I}), t} ||\boldsymbol{\epsilon} - \boldsymbol{\epsilon}_\theta(\sqrt{\alpha_t}\mathbf{x}_0 + (1 - \alpha_t)\boldsymbol{\epsilon}, t)||_2^2. \tag{10}$$

The function $\boldsymbol{\epsilon}_\theta$ estimates noise $\boldsymbol{\epsilon}$ in the noisy input. Once trained, we can sample $\mathbf{x}_0$ from Eq. (2).

# B  Algorithms

## B.1  Algorithm for training and sampling of CSDI

We provide the training procedure of CSDI in Algorithm 1 and the imputation (sampling) procedure with CSDI in Algorithm 2, which are described in Section 4.

---

**Algorithm 1** Training of CSDI

1: **Input:** distribution of training data $q(\mathbf{x}_0)$, a target choice strategy $\mathcal{T}$, the number of iteration $N_{\text{iter}}$, the sequence of noise levels $\{\alpha_t\}$
2: **Output:** Trained denoising function $\boldsymbol{\epsilon}_\theta$
3: **for** $i = 1$ to $N_{\text{iter}}$ **do**
4:     $t \sim \text{Uniform}(\{1, \ldots, T\})$, $\mathbf{x}_0 \sim q(\mathbf{x}_0)$
5:     Separate observed values of $\mathbf{x}_0$ into conditional information $\mathbf{x}_0^{\text{co}}$ and imputation targets $\mathbf{x}_0^{\text{ta}}$ by the target choice strategy $\mathcal{T}$
6:     $\boldsymbol{\epsilon} \sim \mathcal{N}(\mathbf{0}, \mathbf{I})$ where the dimension of $\boldsymbol{\epsilon}$ corresponds to $\mathbf{x}_0^{\text{ta}}$
7:     Calculate noisy targets $\mathbf{x}_t^{\text{ta}} = \sqrt{\alpha_t}\mathbf{x}_0^{\text{ta}} + (1 - \alpha_t)\boldsymbol{\epsilon}$
8:     Take gradient step on $\nabla_\theta ||(\boldsymbol{\epsilon} - \boldsymbol{\epsilon}_\theta(\mathbf{x}_t^{\text{ta}}, t \mid \mathbf{x}_0^{\text{co}}))||_2^2$ according to Eq. (7)

---

**Algorithm 2** Imputation (Sampling) with CSDI

1: **Input:** a data sample $\mathbf{x}_0$, trained denoising function $\boldsymbol{\epsilon}_\theta$
2: **Output:** Imputed missing values $\mathbf{x}_0^{\text{ta}}$
3: Denote observed values of $\mathbf{x}_0$ as $\mathbf{x}_0^{\text{co}}$
4: $\mathbf{x}_T^{\text{ta}} \sim \mathcal{N}(\mathbf{0}, \mathbf{I})$ where the dimension of $\mathbf{x}_T^{\text{ta}}$ corresponds to the missing indices of $\mathbf{x}_0$
5: **for** $t = T$ **to** $1$ **do**
6:     Sample $\mathbf{x}_{t-1}^{\text{ta}}$ using Eq. (5) and Eq. (6)

---

## B.2  Target choice strategies for self-supervised training

We describe the target choice strategies for self-supervised training of CSDI, which is discussed in Section 4.3. We give the algorithm of the random strategy in Algorithm 3 and that of the historical

**Algorithm 3** Target choice with the random strategy

1: **Input:** a training sample $\mathbf{x}_0$
2: **Output:** conditional information $\mathbf{x}_0^{\mathrm{co}}$, imputation targets $\mathbf{x}_0^{\mathrm{ta}}$
3: Draw target ratio $r \sim \mathrm{Uniform}(0, 100)$
4: Randomly choose $r\%$ of the observed values of $\mathbf{x}_0$ and denote the chosen observations as $\mathbf{x}_0^{\mathrm{ta}}$, and denote the remaining observations as $\mathbf{x}_0^{\mathrm{co}}$

---

**Algorithm 4** Target choice with the historical strategy

1: **Input:** a training sample $\mathbf{x}_0$, missing pattern dataset $D_{\mathrm{miss}}$
2: **Output:** conditional information $\mathbf{x}_0^{\mathrm{co}}$, imputation targets $\mathbf{x}_0^{\mathrm{ta}}$
3: Draw a data sample $\tilde{\mathbf{x}}_0$ from $D_{\mathrm{miss}}$
4: Denote the indices of observed values of $\mathbf{x}_0$ as $J$
5: Denote the indices of missing values of $\tilde{\mathbf{x}}_0$ as $\tilde{J}$
6: Take the intersection of $J$ and $\tilde{J}$, and denote values of $\mathbf{x}_0$ for the intersection as $\mathbf{x}_0^{\mathrm{ta}}$
7: Set the remaining observations of $\mathbf{x}_0$ as $\mathbf{x}_0^{\mathrm{co}}$

---

strategy in Algorithm 4. On the historical strategy, we use the training dataset as missing pattern dataset $D_{\mathrm{miss}}$, unless otherwise stated. The mix strategy draws one of the two strategies with a ratio of 1:1 for each training sample. The test pattern strategy just uses the fixed missing pattern in the test dataset to choose imputation targets.

## C  Training and imputation for unconditional diffusion model

### C.1  Imputation with unconditional diffusion model

We describe the imputation method with the unconditional diffusion model used for the experiments in Section 6.1. We followed the method described in previous studies [12]. To utilize unconditional diffusion models for imputation, they approximated the conditional reverse process $p_\theta(\mathbf{x}_{t-1}^{\mathrm{ta}} \mid \mathbf{x}_t^{\mathrm{ta}}, \mathbf{x}_0^{\mathrm{co}})$ in Eq. (5) with the unconditional reverse process in Eq. (2). Given a test sample $\mathbf{x}_0$, they set all observed values as conditional observations $\mathbf{x}_0^{\mathrm{co}}$ and all missing values as imputation targets $\mathbf{x}_0^{\mathrm{ta}}$. Then, instead of conditional observations $\mathbf{x}_0^{\mathrm{co}}$, they considered noisy conditional observations $\mathbf{x}_t^{\mathrm{co}} := \sqrt{\alpha_t}\mathbf{x}_0^{\mathrm{co}} + (1 - \alpha_t)\boldsymbol{\epsilon}$ and exploited $\mathbf{x}_t = [\mathbf{x}_t^{\mathrm{co}}; \mathbf{x}_t^{\mathrm{ta}}] \in \mathcal{X}$ for the input to the distribution $p_\theta(\mathbf{x}_{t-1} \mid \mathbf{x}_t)$ in Eq. (2), where $[\mathbf{x}_t^{\mathrm{co}}; \mathbf{x}_t^{\mathrm{ta}}]$ combines $\mathbf{x}_t^{\mathrm{co}}$ and $\mathbf{x}_t^{\mathrm{ta}}$ to create a sample in $\mathcal{X}$. Using this approximation, we can sample $\mathbf{x}_{t-1}$ from $p_\theta(\mathbf{x}_{t-1} \mid \mathbf{x}_t = [\mathbf{x}_t^{\mathrm{co}}; \mathbf{x}_t^{\mathrm{ta}}])$ and obtain $\mathbf{x}_{t-1}^{\mathrm{ta}}$ by extracting target indices from $\mathbf{x}_{t-1}$. By repeating the sampling procedure from $t = T$ to $t = 1$, we can generate imputation targets $\mathbf{x}_0^{\mathrm{ta}}$.

### C.2  Training procedure of unconditional diffusion models for time series imputation

In Section 3.2, we described the training procedure of the unconditional diffusion model, which expects the training dataset does not contain missing values. However, the training dataset that we use for time series imputation contains missing values. To handle missing values, we slightly modify the training procedure. Given a training sample $\mathbf{x}_0$ with missing values, we treat the missing values like observed values by filling dummy values to the missing indices of $\mathbf{x}_0$. We adopt zeros for the dummy values and denote the training sample after filling zeros as $\widehat{\mathbf{x}}_0$. Since all indices of $\widehat{\mathbf{x}}_0$ contain values, we can sample noisy targets $\sqrt{\alpha_t}\widehat{\mathbf{x}}_0 + (1 - \alpha_t)\boldsymbol{\epsilon}$ as with the training procedure in Section 3.2. We consider denoising the noisy target for training, but we are only interested in estimating the noises added to the observed indices since the dummy values contain no information about the data distribution. To exclude the missing indices, we introduce an observation mask $\mathbf{m} \in \{0, 1\}^{K \times L}$, which takes value 1 for observed indices. Then, instead of Eq. (4), we use the following loss function for training under the existence of missing values:

$$\min_\theta L(\theta) := \min_\theta \mathbb{E}_{\mathbf{x}_0 \sim q(\mathbf{x}_0), \boldsymbol{\epsilon} \sim \mathcal{N}(\mathbf{0}, \mathbf{I}), t} ||(\boldsymbol{\epsilon} - \boldsymbol{\epsilon}_\theta(\sqrt{\alpha_t}\widehat{\mathbf{x}}_0 + (1 - \alpha_t)\boldsymbol{\epsilon}, t)) \odot \mathbf{m}||_2^2. \quad (11)$$

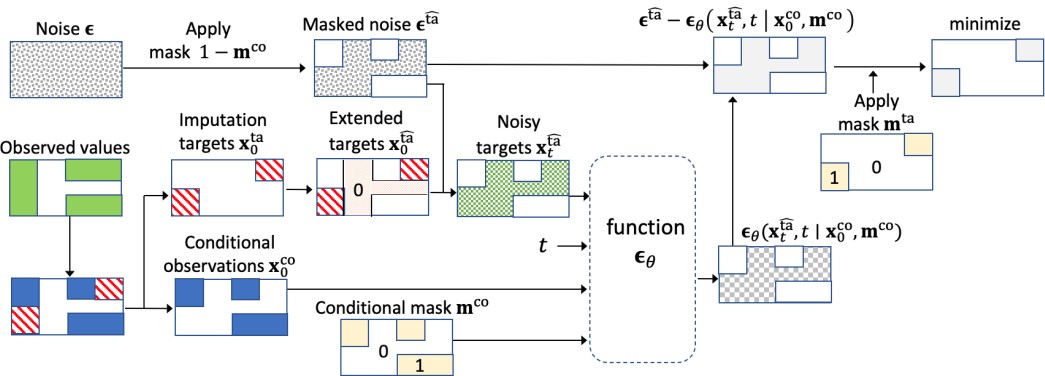

Figure 5: The self-supervised training procedure of CSDI for implementation of time series imputation. The colored areas in each rectangle represent the existence of values. The green and white areas represent observed and missing values, respectively, and white areas are padded with zeros to fix the shape of the inputs. Zero padding is also applied to all white areas. As with Figure 2, the observed values are separated into red imputation targets $\mathbf{x}_0^{\text{ta}}$ and blue conditional observations $\mathbf{x}_0^{\text{co}}$. For the extended targets $\mathbf{x}_0^{\widehat{\text{ta}}}$, the area of value 0 shows dummy values.

## D    CSDI for implementation of time series imputation

In this section, we discuss the effect of adjusting the function $\epsilon_\theta$, described in Section 5. First, let us consider the effect of the adjustment on sampling. The adjustment does not essentially affect the model at sampling time, because all values of data are either conditional observations or imputation targets as shown in Table 1 and the model can distinguish the type of each value through the mask $\mathbf{m}^{\text{co}}$. Since the output shape is adjusted, we need to recover the shape by extracting the indices of the imputation targets from the output, so that we substitute the outputs into Eq. (6).

Next, we focus on the effect of the adjustment on training. Unlike sampling, the model at training time cannot distinguish imputation targets and missing values since we ignore missing values during training as shown in Table 1. In order to handle the missing values, we need to modify the inputs to $\epsilon_\theta$. Here, we use a similar approach to the training procedure with the unconditional model in Section C.2. Namely, we treat the missing indices like a part of imputation targets. We illustrate the extended training procedure in Figure 5. First, we set zeros to the missing indices as dummy values. We denote the extended imputation targets as $\mathbf{x}_0^{\widehat{\text{ta}}}$. Then, we sample noisy targets $\mathbf{x}_t^{\widehat{\text{ta}}} = \sqrt{\alpha_t}\mathbf{x}_0^{\widehat{\text{ta}}} + (1 - \alpha_t)\boldsymbol{\epsilon}^{\widehat{\text{ta}}}$, where $\boldsymbol{\epsilon}^{\widehat{\text{ta}}}$ is masked noise and is given by $\boldsymbol{\epsilon}^{\widehat{\text{ta}}} := (1 - \mathbf{m}^{\text{co}}) \odot \boldsymbol{\epsilon}$, as shown in Figure 5. We denoise the noisy targets for training. We only estimate the noise for the original imputation targets, since the dummy values contain no information about the data distribution. In other words, we train $\epsilon_\theta$ by solving the following optimization problem:

$$\min_\theta \mathcal{L}(\theta) := \min_\theta \mathbb{E}_{\mathbf{x}_0 \sim q(\mathbf{x}_0), \boldsymbol{\epsilon} \sim \mathcal{N}(\mathbf{0},\mathbf{I}),t} ||(\boldsymbol{\epsilon} - \epsilon_\theta(\mathbf{x}_t^{\widehat{\text{ta}}}, t \mid \mathbf{x}_0^{\text{co}}, \mathbf{m}^{\text{co}})) \odot \mathbf{m}^{\text{ta}}||_2^2 \qquad (12)$$

where $\mathbf{m}^{\text{ta}}$ is a mask which corresponds to $\mathbf{x}_0^{\text{ta}}$ and takes value 1 for the original imputation targets.

## E    Details of architectures and experiment settings

### E.1    Details of implementation of CSDI

We describe the details of architectures and hyperparameters for the conditional diffusion model described in Section 5. First, we provide the whole architecture of CSDI in Figure 6. Since the architecture in Figure 6 is based on DiffWave [13], we mainly explain the difference from DiffWave.

On the top of the figure, the models take $\mathbf{x}_0^{\text{co}}$ and $\mathbf{x}_t^{\text{ta}}$ as inputs since $\epsilon_\theta$ is the conditional denoising function. For the diffusion step $t$, we use the following 128-dimensions embedding following previous works [29, 13]:

$$t_{embedding}(t) = \left( \sin(10^{0\cdot4/63}t), \dots, \sin(10^{63\cdot4/63}t), \cos(10^{0\cdot4/63}t), \dots, \cos(10^{63\cdot4/63}t) \right). \quad (13)$$

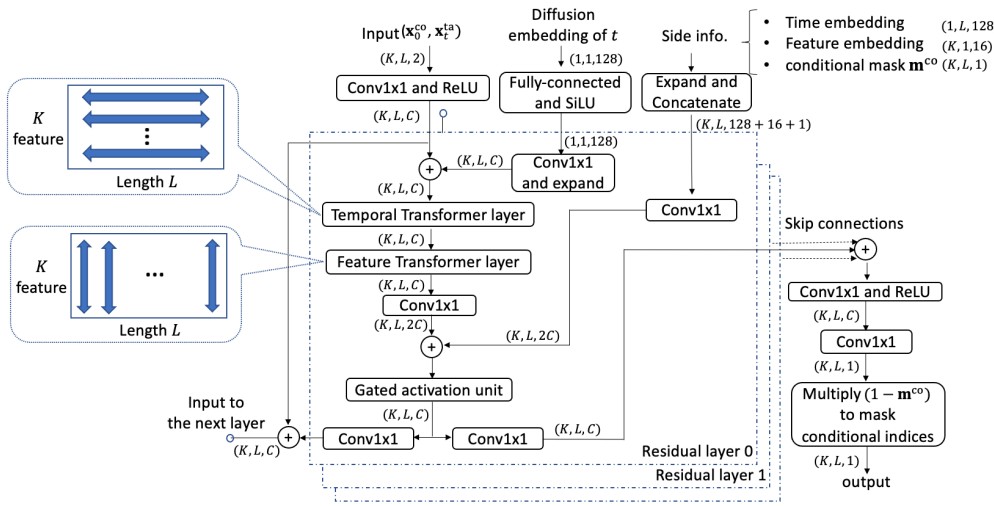

Figure 6: Architecture of $\epsilon_\theta$ in CSDI for multivariate time series imputation.

Similarly, we utilize time embedding of $\mathbf{s} = \{s_{1:L}\}$ as a side information. We use 128-dimensions temporal embedding following previous studies [29, 30]:

$$s_{embedding}(s_l) = \left( \sin(s_l/\tau^{0/64}), \ldots, \sin(s_l/\tau^{63/64}), \cos(s_l/\tau^{0/64}), \ldots, \cos(s_l/\tau^{63/64}) \right) \quad (14)$$

where $\tau = 10000$. On the top right of the figure, we expand each side information and concatenate all the side information. On the bottom right of the figure, we multiply the output by a mask $(1 - \mathbf{m}^{co})$ in order to mask the indices of the conditional observations of the output.

As for Transformer layers, we used 1-layer TransformerEncoder implemented in PyTorch [39], which is composed of a multi-head attention layer, fully-connected layers and layer normalization. Only for forecasting tasks, we adopted the "linear attention transformer" package [40] to improve computational efficiency, since the forecasting datasets we used contained many features and long sequences. The package implements an efficient attention mechanism [41], and we only used global attention in the package.

## E.2 Details of experiment settings in Section 6

In this section, we provide the details of the experiment settings in Section 6. When we evaluated baseline methods with the original implementation in each section, we used their original hyperparameters and model size. Although we also ran experiments under the same model size as our model, the performance did not improve in more than half of the cases and did not outperform our model in all cases.

### E.2.1 Experiment settings for imputation in Section 6.1

First, we explain additional information for the air quality dataset. The dataset is composed of air quality data in Beijing from 2014/05/01 to 2015/04/30. The dataset contains artificial ground-truth, whose missing patterns are created based on those in the next month.

Next, we describe data splits. For the healthcare dataset, we randomly divided the dataset into five parts and used one of them as test data for each run. We also randomly split the remaining data into train and validation data with a ratio of 7:1. For the air quality dataset, following [2], we used the 3rd, 6th, 9th and 12th months as test data. To avoid evaluating imputation for each missing value multiple times, we separated the test data of each month every 36 consecutive time steps without overlap. When the length of a monthly data was not divisible by 36, we allowed the last sequence to overlap with the previous one and did not aggregate the result for the overlapping parts. For each run, we selected a month as validation data and used the rest as training data. We note that we excluded the 4th, 7th, 10th, and 1st months from missing pattern dataset for the historical strategy, because these months were used for creating missing patterns of the artificial ground-truth.

On the healthcare dataset, due to the different scales of features, we evaluate the performance on normalized data following previous studies [7]. For training of all tasks, we normalize each feature to have zero mean and unit variance.

As for hyperparameters, we set the batch size as 16 and the number of epochs as 200. We used Adam optimizer with learning rate 0.001 that is decayed to 0.0001 and 0.00001 at 75% and 90% of the total epochs, respectively. As for the model, we set the number of residual layers as 4, residual channels as 64, and attention heads as 8. We followed DiffWave[13] for the number of channels and decided the number of layers based on the validation loss and the parameter size. The number of the parameter in the model is about 415,000.

We also provide hyperparameters for the diffusion model as follows. We set the number of the diffusion step $T = 50$, the minimum noise level $\beta_1 = 0.0001$, and the maximum noise level $\beta_T = 0.5$. Since recent studies[38, 42] reported that gentle decay of $\alpha_t$ could improve the sample quality, we adopted the following quadratic schedule for other noise levels:

$$\beta_t = \left( \frac{T-t}{T-1} \sqrt{\beta_1} + \frac{t-1}{T-1} \sqrt{\beta_T} \right)^2 . \tag{15}$$

With regard to the baselines for probabilistic imputation, we used their original implementations for GP-VAE and V-RIN. For Multitask GP, we utilized GPyTorch [43] for the implementation. We used RBF kernel for the covariance between timepoints and low-rank IndexKernel with $rank = 10$ for that between features.

Finally, we describe the baselines for deterministic imputation, which were used for comparison. 1) BRITS [7]: the method utilizes a bi-directional recurrent neural network to handle multiple correlated missing values. 2) V-RIN [32]: the method utilizes the uncertainty learned with VAE to improve recurrent imputation. 3) GLIMA [21]: the method combines recurrent imputations with an attention mechanism to capture cross-time and cross-feature dependencies and shows the state-of-the-art performance. 4) RDIS [20]: the method applies random drops to given training data for self-training. We used the original implementation for BRITS and V-RIN. For RDIS, we set the number of models as 8, hidden units as 108, drop rate as 30%, threshold as 0.1, update epoch as 200, and total epochs as 1000.

### E.2.2 Experiment settings for interpolation in Section 6.2

First, we explain how we process the dataset. We processed the healthcare dataset as irregularly sampled time series. Following previous studies [22, 35], we rounded observation times to the nearest minute. Then, there are $48 \times 60$ possible measurement times per time series, and the lengths of time series samples can be different each other.

We used almost the same experiment settings as those for imputation in Section E.2.1. Since the length of each irregularly sampled time series is different, we applied zero padding to each time series in order to fix the length for each batch. The padding does not affect the result since the attention mechanisms in the implementation of CSDI can deal with the padding by using padding masks.

We describe the baselines which were used for comparison. We used the original implementation. 1) Latent ODE [35]: the method consists of an ODE-RNN model as the encoder and a neural ODE model as the decoder. 2) mTANs [22]: the method utilized an attention mechanism and showed state-of-the-art results for the interpolation of irregularly sampled time series.

Table 6: Description of datasets for time series forecasting.

| | feature $K$ | total time step | history steps $L_1$ | prediction steps $L_2$ | test sample | epochs |
|---|---|---|---|---|---|---|
| solar | 137 | 10392 | 168 | 24 | 7 | 50 |
| electricity | 370 | 5833 | 168 | 24 | 7 | 100 |
| traffic | 963 | 7009 | 168 | 24 | 7 | 200 |
| taxi | 1214 | 1488 | 48 | 24 | 56 | 300 |
| wiki | 2000 | 792 | 90 | 30 | 5 | 300 |

### E.2.3   Datasets and Experiment settings for forecasting in Section 6.3

First we describe the datasets we used. We used five open datasets that are commonly used for evaluating probabilistic time series forecasting. The datasets were preprocessed in Salinas et al. [34] and provided in GluonTS[1][44]:

- solar [45]: hourly solar power production records of 137 stations in Alabama State.
- electricity[2]: hourly electricity consumption of 370 customers.
- traffic[3]: hourly occupancy rate of 963 San Fancisco freeway car lanes.
- taxi[4]: half hourly traffic time series of New York taxi rides taken at 1214 locations in the months of January 2015 for training and January 2016 for test.
- wiki: daily page views of 2000 Wikipedia pages.

We summarize the characteristics of each dataset in Table 6. The task for these datasets is to predict the future $L_2$ steps by exploiting the latest $L_1$ steps where $L_1$ and $L_2$ depend on datasets as shown in Table 6. We set $L_1$ and $L_2$ referring to previous studies [37]. For training, we randomly selected $L_1 + L_2$ consecutive time steps as one time series and set the last $L_2$ steps as imputation targets. We followed the train/test split in previous studies. We used the last five samples of training data as validation data.

As for experiment settings, since we basically followed the setting for time series imputation in Section E.2.1, we only describe the difference from it. We ran each experiment three times with different random seeds. We set batch size as 8 because of longer sequence length, and utilized an efficient Transformer as mentioned in Section E.1.

Since the number of features $K$ is large, we adopted subset sampling of features for training. For each time series in a training batch, we randomly chose a subset of features and only used the subset for the batch. The attention mechanism allows the model to take varying length inputs. We set the subset size as $64$. Due to the subset sampling, we need large epochs when the number of features $K$ is large. Therefore, we set training epochs based on the number of features and the validation loss. We provide the epochs in Table 6.

Finally, we describe the baselines which were used for comparison. 1) GP-copula [34]: the method combines a RNN-based model with a Gaussian copula process to model time-varying correlations. 2) Transformer MAF [36]: the method uses Transformer to learn temporal dynamics and a conditioned normalizing flow to capture feature dependencies. 3) TLAE [37]: the method combines a RNN-based model with autoencoders to learn latent temporal patterns. 4) TimeGrad [25]: the method has shown the state-of-the-art results for probabilistic forecasting by combining a RNN-based model with diffusion models.

### E.3   Computations of CRPS

We describe the definition and computation of the CRPS metric.

The continuous ranked probability score (CRPS) [33] measures the compatibility of an estimated probability distribution $F$ with an observation $x$, and can be defined as the integral of the quantile loss $\Lambda_\alpha(q, z) := (\alpha - \mathbb{1}_{z<q})(z - q)$ for all quantile levels $\alpha \in [0, 1]$:

$$\mathrm{CRPS}(F^{-1}, x) = \int_0^1 2\Lambda_\alpha(F^{-1}(\alpha), x)\mathrm{d}\alpha \tag{16}$$

where $\mathbb{1}$ is the indicator function. We generated 100 samples to approximate the distribution $F$ over each missing value. We computed quantile losses for discretized quantile levels with $0.05$ ticks. Namely, we approximated CRPS with

$$\mathrm{CRPS}(F^{-1}, x) \simeq \sum_{i=1}^{19} 2\Lambda_{i*0.05}(F^{-1}(i * 0.05), x)/19. \tag{17}$$

---

[1]https://github.com/awslabs/gluon-ts
[2]https://archive.ics.uci.edu/ml/datasets/ElectricityLoadDiagrams20112014
[3]https://archive.ics.uci.edu/ml/datasets/PEMS-SF
[4]https://www1.nyc.gov/site/tlc/about/tlc-trip-record-data

Then, we evaluated the following normalized average of CRPS for all features and time steps:

$$\frac{\sum_{k,l} \text{CRPS}(F_{k,l}^{-1}, x_{k,l})}{\sum_{k,l} |x_{k,l}|} \tag{18}$$

where $k$ and $l$ indicates features and time steps of imputation targets, respectively.

For probabilistic forecasting, we evaluated CRPS-sum. CRPS-sum is CRPS for the distribution $F$ of the sum of all $K$ features and is computed by the following equation:

$$\frac{\sum_{l} \text{CRPS}(F^{-1}, \sum_{k} x_{k,l})}{\sum_{k,l} |x_{k,l}|} \tag{19}$$

where $\sum_{k} x_{k,l}$ is the sum of forecasting targets for all features at time point $l$.

## F  Additional results and experiments

Table 7: Comparing the two dimension attention mechanism of various architectures. For ablations, we report the mean and the standard error for three trials.

|  | healthcare (10% missing) | | air quality | |
|---|---|---|---|---|
|  | MAE | CRPS | MAE | CRPS |
| no-temporal | 0.439(0.004) | 0.475(0.001) | 26.63(0.23) | 0.292(0.002) |
| no-feature | 0.352(0.001) | 0.386(0.002) | 14.44(0.11) | 0.162(0.001) |
| flatten | 0.383(0.002) | 0.418(0.002) | 12.26(0.09) | 0.139(0.001) |
| Bi-RNN | 0.272(0.001) | 0.301(0.001) | 12.56(0.26) | 0.142(0.003) |
| dilated conv | 0.279(0.002) | 0.305(0.002) | 11.67(0.11) | 0.130(0.001) |
| 2D attention (proposed) | **0.217**(**0.001**) | **0.238**(**0.001**) | **9.60**(**0.04**) | **0.108**(**0.001**) |

### F.1  Effectiveness of two dimensional attention mechanism

In this paper, we utilized a two dimensional attention mechanism to learn temporal and feature dependencies. To show the effectiveness of the attention mechanism, we demonstrate an ablation study. We replace the attention mechanism with the following architecture baselines and compare the performance:

- no temporal: remove temporal attention layers

- no feature: remove feature attention layers

- flatten: flatten 2D tensor ($K$ features x $L$ length) to 1D, and input the 1D vector to transformer layers

- Bi-RNN: replace the attention mechanism with Bi-directional RNN which is a popular architecture for multivariate time series imputation

- dilated conv: replace temporal and feature attention layers with 1D dilated convolution layers, respectively. The dilated convolution was used in previous studies for diffusion models[13, 25]

We set hyperparameters of each architecture so that the number of parameters is almost the same as our attention mechanism. We show the result in Table 7. Our attention mechanism outperforms all of the other architectures. The comparison with "no temporal" and "no feature" shows that both temporal and feature correlations are important for accurate imputation. The comparison with "flatten", "Bi-RNN", and "dilated conv" shows that our attention mechanism is effective to learn temporal and feature dependency compared with existing methods. In summary, the result of the ablation indicates the proposed attention mechanism plays a key role in improving the imputation performance by a large margin.

Table 8: Comparison of the negative log likelihood (NLL) and CRPS for various schedules. We report the mean for three trials.

| method | schedule | healthcare (10% missing) | | air quality | |
| | | NLL | CRPS | NLL | CRPS |
|---|---|---|---|---|---|
| GP-VAE | – | $< 1.22$ | 0.574 | $< 1.09$ | 0.397 |
| proposed | quad. (in paper) | $< 1.63$ | **0.238** | $< 0.97$ | **0.108** |
| proposed | linear | $< 29.70$ | 0.240 | $< 18.55$ | 0.110 |
| proposed | quad. (large min. noise) | $< \mathbf{0.07}$ | 0.239 | $< \mathbf{-0.70}$ | 0.109 |

### F.2 Comparison of negative log likelihood for probabilistic imputation

The negative log likelihood (NLL) is a popular metric for evaluating probabilistic methods and ELBO is often utilized to estimate NLL. A reason why we mainly focused on other metrics is that ELBO is sometimes far from NLL and uncorrelated with the quality of generated samples. Specifically, in the proposed method, the choice of the noise schedule highly affects the ELBO while it has little effect on the sample quality.

To demonstrate this, we performed an experiment. We chose the following three noise schedules for CSDI and calculated NLL and CRPS for each schedule.

- quadratic (used in the paper): quadratic spaced schedule between $\beta_{\min} = 0.0001$ and $\beta_{\max} = 0.5$
- linear: linear spaced schedule with the same $\beta_{\min}$ and $\beta_{\max}$ as those in the paper
- quadratic (large minimum noise): quadratic schedule with large minimum noise level $\beta_{\min} = 0.001$, which makes the model ignore small noise

We also calculated the metrics for GP-VAE. The result is shown in Table 8. While CRPS by the proposed method is almost independent from the choice of schedules, NLL significantly depends on the schedule. This phenomenon happens because time series data is generally noisy and it is difficult to denoise small noise during imputation. Estimated scores by the model could be inaccurate when inputs to the model (i.e. imputation targets) only contain small noise. These inaccurate scores could make the estimated ELBO loose, whereas small noise does not affect the sample quality. When the minimum noise level $\beta_{\min}$ is large, since the model does not denoise small noise in sampling steps, ELBO by the proposed method is tightly estimated and smaller than that by GP-VAE. Therefore, ELBO is not suitable for evaluating the sample quality and we adopted other metrics such as CRPS and MAE.

### F.3 Experimental results for other metrics in Section 6

We show the experimental results in Section 6 for different metrics in Table 9 to 12. Table 9 evaluates RMSE for deterministic imputation methods. We added SSGAN [19] as an additional baseline, which has shown the state-of-the-art performance for RMSE in the healthcare dataset. We can confirm that CSDI outperforms all baselines for RMSE. The advantage of CSDI is particularly large when the missing ratio is low. This result is consistent with that in Section 6.1.

Table 10 evaluates MAE and RMSE for interpolation methods. The result is consistent with Table 4. Table 11 and 12 report CRPS and MSE for probabilistic forecasting methods, respectively. We exclude TimeGrad [25] from the baselines, as they did not report these metrics. We can see that CSDI is competitive with baselines for these metrics as with CRPS-sum.

### F.4 Effect of the number of generated samples

For the experiments in Section 6, we generated 100 samples to estimate the distribution of imputation. We demonstrate the relationship between the number of samples and the performance in Figure 7. We can see that five or ten samples are enough to estimate good distributions and outperform the baselines. Increasing the number of samples further improves the performance, and the improvement becomes marginal over 50 samples.

Table 9: Comparing deterministic imputation methods with CSDI for RMSE. The results correspond to Table 3. We report the mean and the standard error for five trials. The asterisk means the values are cited from the original paper.

| | healthcare | | | air quality |
| --- | --- | --- | --- | --- |
| | 10% missing | 50% missing | 90% missing | |
| V-RIN [32] | 0.628(0.025) | 0.693(0.022) | 0.928(0.013) | 40.11(1.14) |
| BRITS [7] | 0.619(0.022) | 0.693(0.023) | 0.836(0.015) | 24.47(0.73) |
| RDIS [20] | 0.633(0.021) | 0.741(0.018) | 0.934(0.013) | 37.49(0.28) |
| SSGAN [19] (*) | 0.598 | 0.762 | 0.818 | − |
| unconditional | 0.621(0.020) | 0.734(0.024) | 0.940(0.018) | 22.58(0.23) |
| **CSDI** (proposed) | **0.498(0.020)** | **0.614(0.017)** | **0.803(0.012)** | **19.30(0.13)** |

Table 10: Comparing the state-of-the-art interpolation method with CSDI for MAE and RMSE. The results correspond to Table 4. We report the mean and the standard error for five trials.

| | | 10% missing | 50% missing | 90% missing |
| --- | --- | --- | --- | --- |
| MAE | Latent ODE [35] | 0.522(0.002) | 0.506(0.003) | 0.578(0.009) |
| | mTANs [22] | 0.389(0.003) | 0.422(0.003) | 0.533(0.005) |
| | **CSDI** (proposed) | **0.362(0.001)** | **0.394(0.002)** | **0.518(0.003)** |
| RMSE | Latent ODE [35] | 0.799(0.012) | 0.783(0.012) | 0.865(0.017) |
| | mTANs [22] | 0.749(0.037) | 0.721(0.014) | **0.836(0.018)** |
| | **CSDI** (proposed) | **0.722(0.043)** | **0.700(0.013)** | 0.839(0.009) |

Table 11: Comparing probabilistic forecasting methods with CSDI for CRPS. The results correspond to Table 5. We report the mean and the standard error for three trials. The results for baseline methods are cited from their paper. 'TransMAF' is the abbreviation for 'Transformer MAF'.

| | solar | electricity | traffic | taxi | wiki |
| --- | --- | --- | --- | --- | --- |
| GP-copula [34] | 0.371(0.022) | 0.056(0.002) | 0.133(0.001) | 0.360(0.201) | 0.236(0.000) |
| TransMAF [36] | 0.368(0.001) | 0.052(0.000) | 0.134(0.001) | 0.377(0.002) | 0.274(0.007) |
| TLAE [37] | **0.335(0.025)** | 0.058(0.002) | 0.097(0.001) | 0.369(0.006) | 0.298(0.001) |
| **CSDI** (proposed) | 0.338(0.012) | **0.041(0.000)** | **0.073(0.000)** | **0.271(0.001)** | **0.207(0.002)** |

Table 12: Comparing probabilistic forecasting methods with CSDI for MSE. The results correspond to Table 5. We report the mean and the standard error for three trials. The results for baseline methods are cited from their paper. 'TransMAF' is the abbreviation for 'Transformer MAF'. 'TransMAF' did not report the standard error.

| | solar | electricity | traffic | taxi | wiki |
| --- | --- | --- | --- | --- | --- |
| GP-copula [34] | 9.8e2(5.2e1) | 2.4e5(5.5e4) | 6.9e-4(2.2e-5) | 3.1e1(1.4e0) | 4.0e7(1.6e9) |
| TransMAF [36] | 9.3e2 | 2.0e5 | 5.0e-4 | 4.5e1 | **3.1e7** |
| TLAE [37] | **6.8e2(7.5e1)** | 2.0e5(9.2e4) | 4.0e-4(2.9e-6) | 2.6e1(8.1e-1) | 3.8e7(4.2e4) |
| **CSDI** (proposed) | 9.0e2(6.1e1) | **1.1e5(2.8e3)** | **3.5e-4(7.0e-7)** | **1.7e1(6.8e-2)** | 3.5e7(4.4e4) |

Table 13: The effect of the target choice strategy for the air quality dataset. We report the mean and the standard error for five trials.

| | CRPS | MAE |
| --- | --- | --- |
| random | **0.108(0.001)** | **9.58(0.08)** |
| historical | 0.113(0.001) | 10.12(0.05) |
| mix | **0.108(0.001)** | 9.60(0.04) |

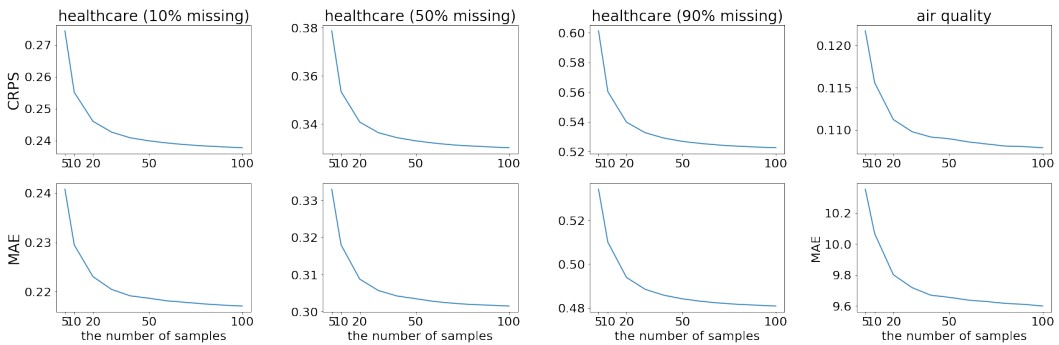

Figure 7: The effect of the number of generated samples. The first row shows the effect on probabilistic imputation in Table 2 and the second row shows the effect on deterministic imputation in Table 3.

## F.5 Effect of target choice strategy

In the experiment for the air quality dataset in Section 6.1, we adopted the mix strategy for the target choice. Here, we provide the result for other strategies and show the effect of the target choice strategy on imputation quality. In Table 13, the performances of the mix strategy and the random strategy are almost the same, and the performance of the historical strategy is slightly worse than that of the other strategies. This means that the historical strategy is not effective for the air quality dataset even though the dataset contains structured missing patterns. This is due to the difference of missing patterns between training dataset and test dataset. Note that all strategies outperform the baselines in Table 2 and Table 3.

## G Additional examples of probabilistic imputation

In this section, we illustrate various imputation examples to show the characteristic of imputed samples. We pick a multivariate time series from the results of each experiment in Section 6.1 and show imputation results for all features of each time series in Figure 8 to 11. We compare CSDI with GP-VAE in Figure 8 to 11. Note that the scales of the $y$ axis depend on the features. For the healthcare dataset with 90% missing ratio in Figure 10, while GP-VAE fails to learn the distribution, CSDI gives reasonable probabilistic imputation for most of the features. For the air quality dataset in Figure 11, CSDI learns the dependency between features and provides more accurate imputation than GP-VAE. In Figure 12 to 15, we compare CSDI with the unconditional diffusion model. In all figures, CSDI tends to provide tighter uncertainty than the unconditional diffusion model. We hypothesize that it is due to the approximation discussed in Section 3.3. Since the unconditional model approximates the conditional distribution by using noisy observed values, the estimated imputation become less confident than that with the conditional model.

## H Potential negative societal impacts

Since score-based diffusion models are generative models, our proposed model has negative impacts as well as other generative models. For example, the model can potentially memorize private information and be used to generate fake data.

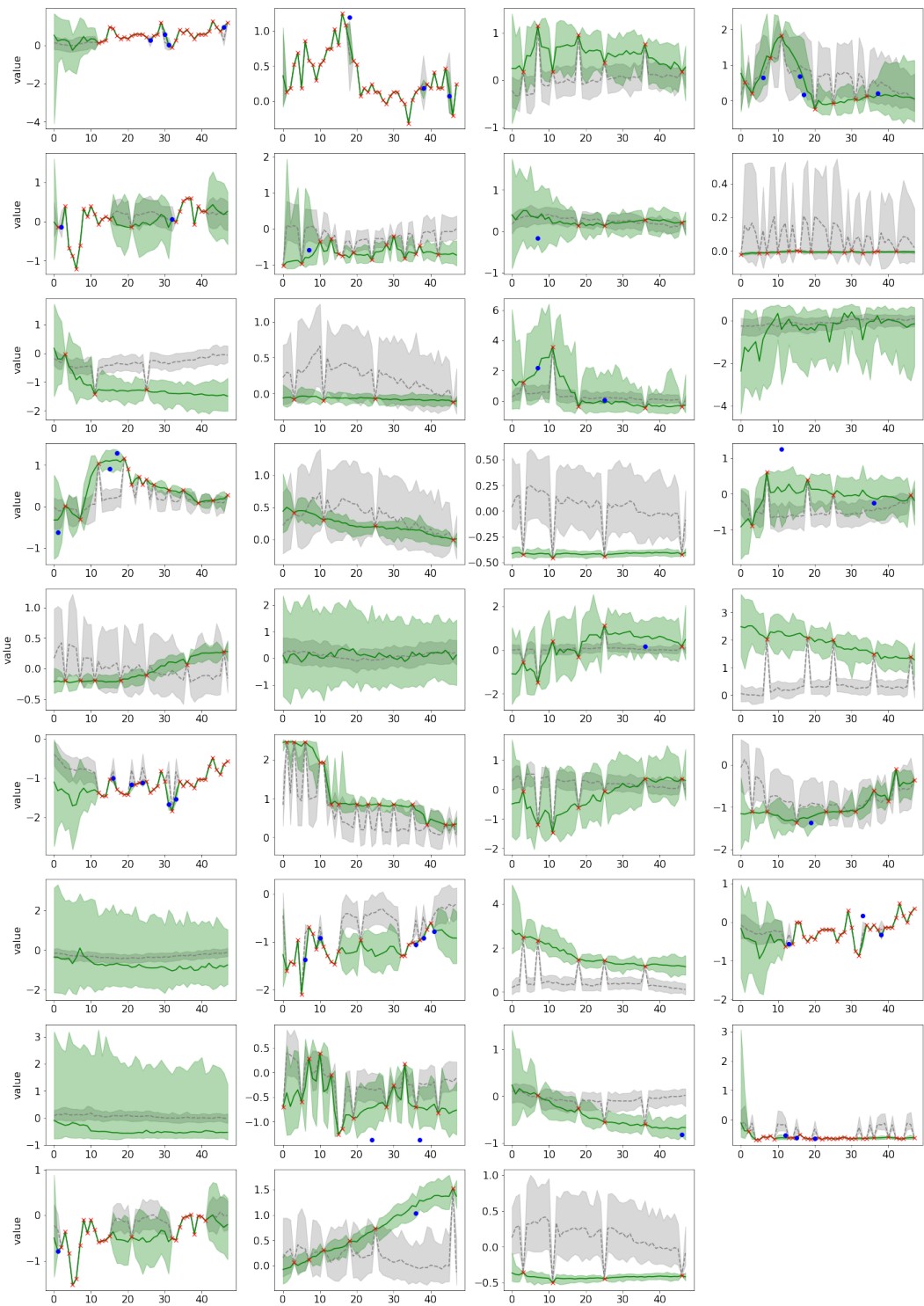

Figure 8: Comparison of imputation between GP-VAE and CSDI for the healthcare dataset (10% missing). The result is for a time series sample with all 35 features. The red crosses show observed values and the blue circles show ground-truth imputation targets. Green and gray colors correspond to CSDI and GP-VAE, respectively. For each method, median values of imputations are shown as the line and 5% and 95% quantiles are shown as the shade.

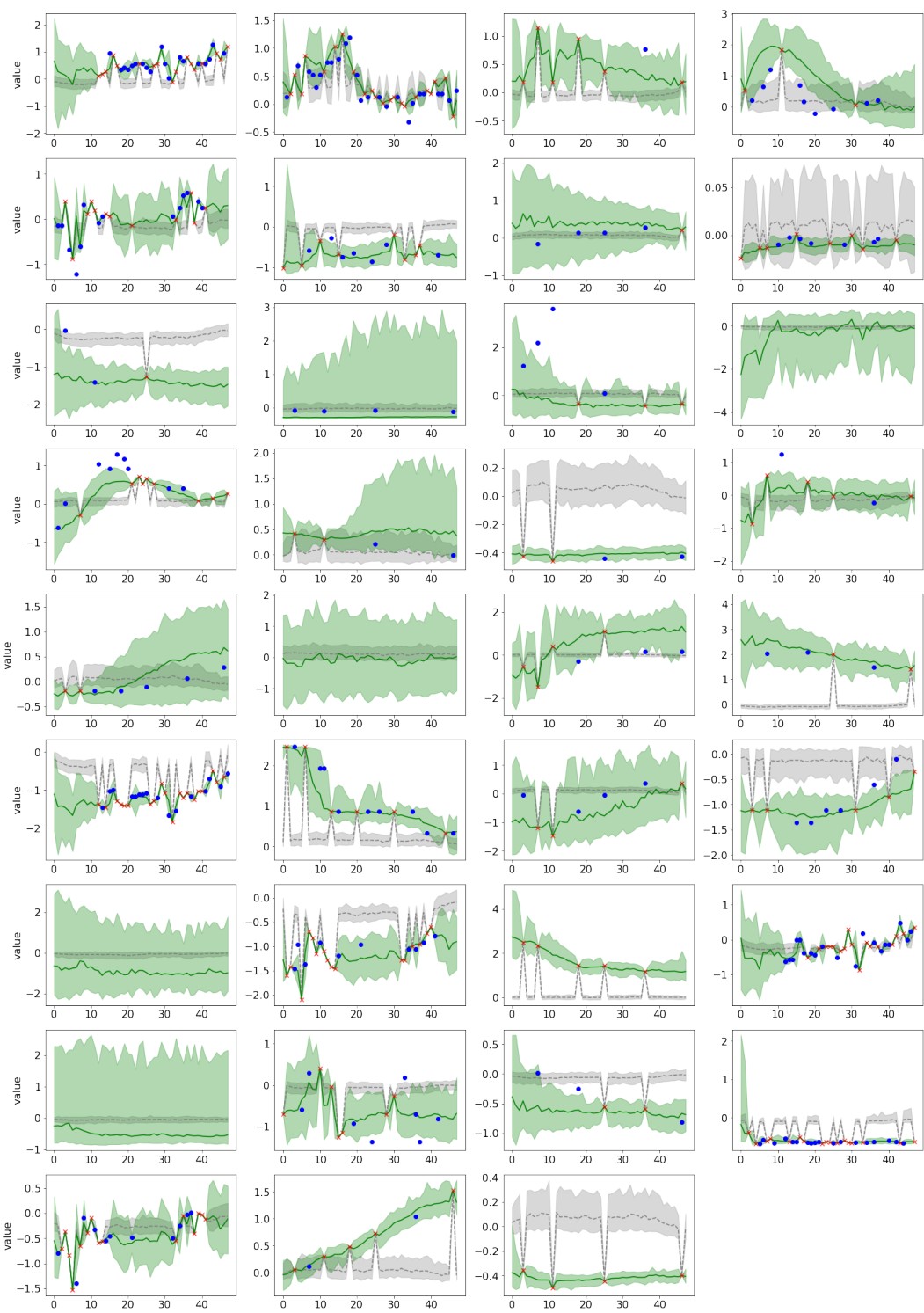

Figure 9: Comparison of imputation between GP-VAE and CSDI for the healthcare dataset (50% missing). The result is for a time series sample with all 35 features. The red crosses show observed values and the blue circles show ground-truth imputation targets. Green and gray colors correspond to CSDIand GP-VAE, respectively. For each method, median values of imputations are shown as the line and 5% and 95% quantiles are shown as the shade.

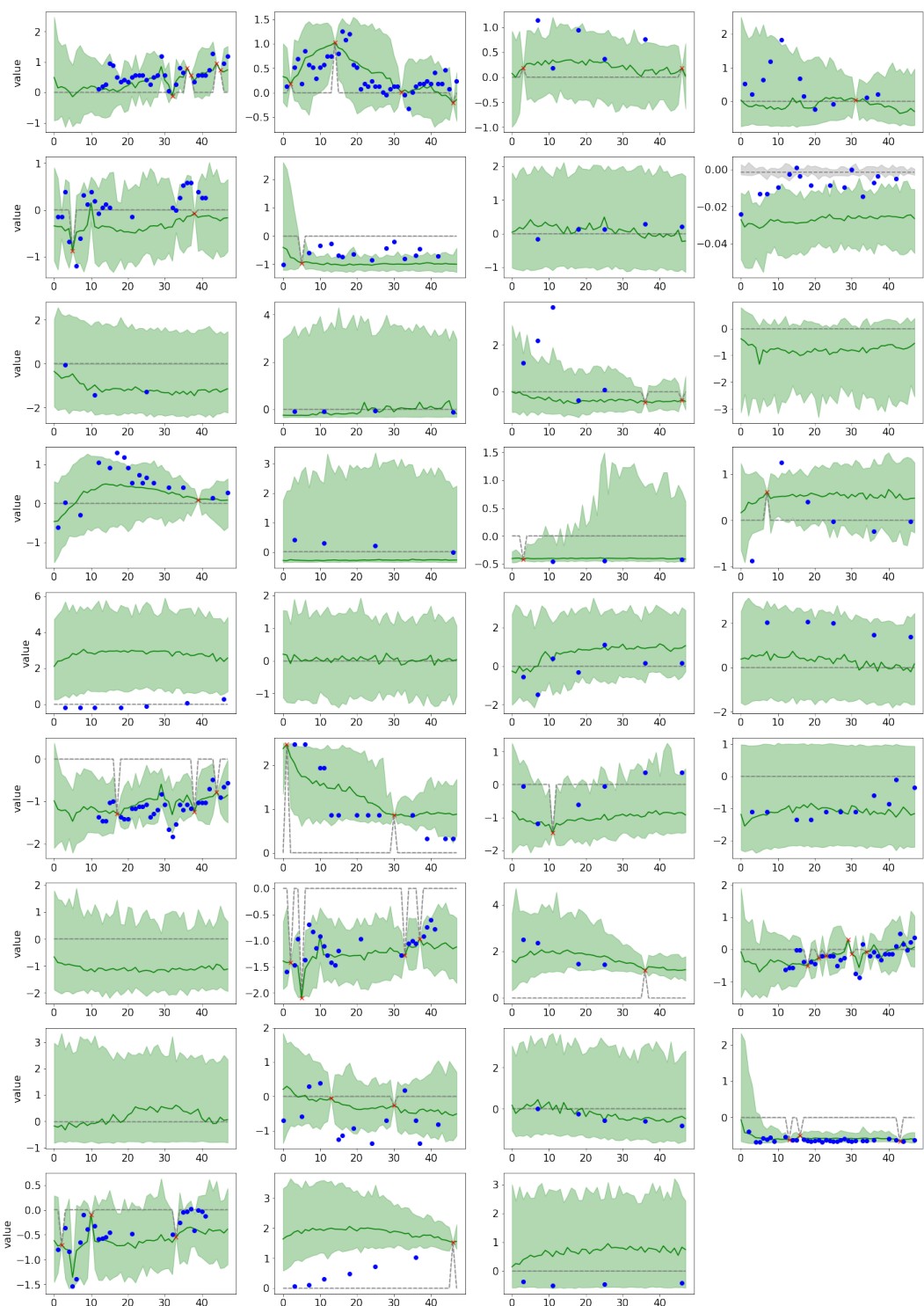

Figure 10: Comparison of imputation between GP-VAE and CSDI for the healthcare dataset (90% missing). The result is for a time series sample with all 35 features. The red crosses show observed values and the blue circles show ground-truth imputation targets. Green and gray colors correspond to CSDI and GP-VAE, respectively. For each method, median values of imputations are shown as the line and 5% and 95% quantiles are shown as the shade.

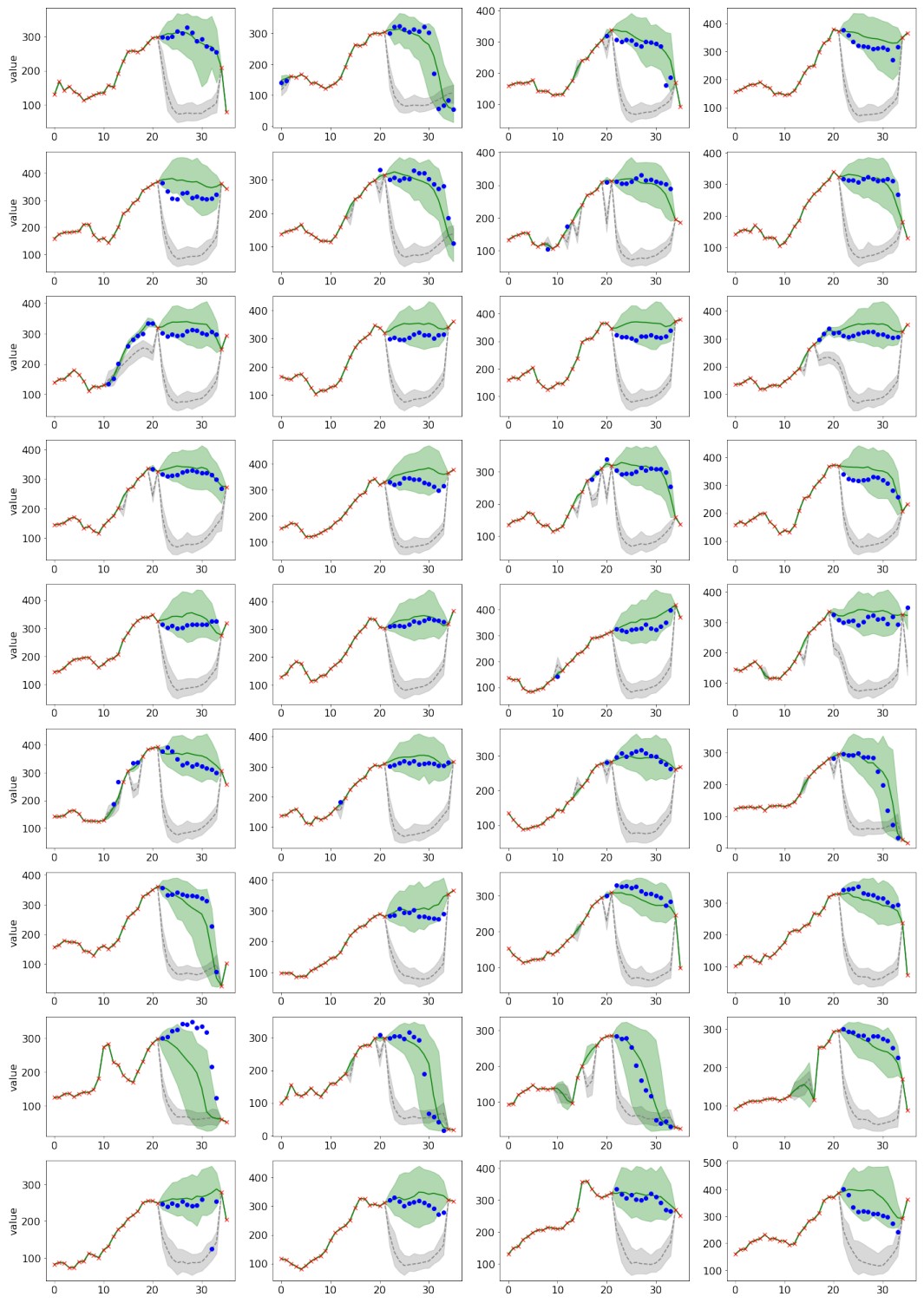

Figure 11: Comparison of imputation between GP-VAE and CSDI for the air quality dataset. The result is for a time series sample with all 36 features. The red crosses show observed values and the blue circles show ground-truth imputation targets. Green and gray colors correspond to CSDI and GP-VAE, respectively. For each method, median values of imputations are shown as the line and 5% and 95% quantiles are shown as the shade.

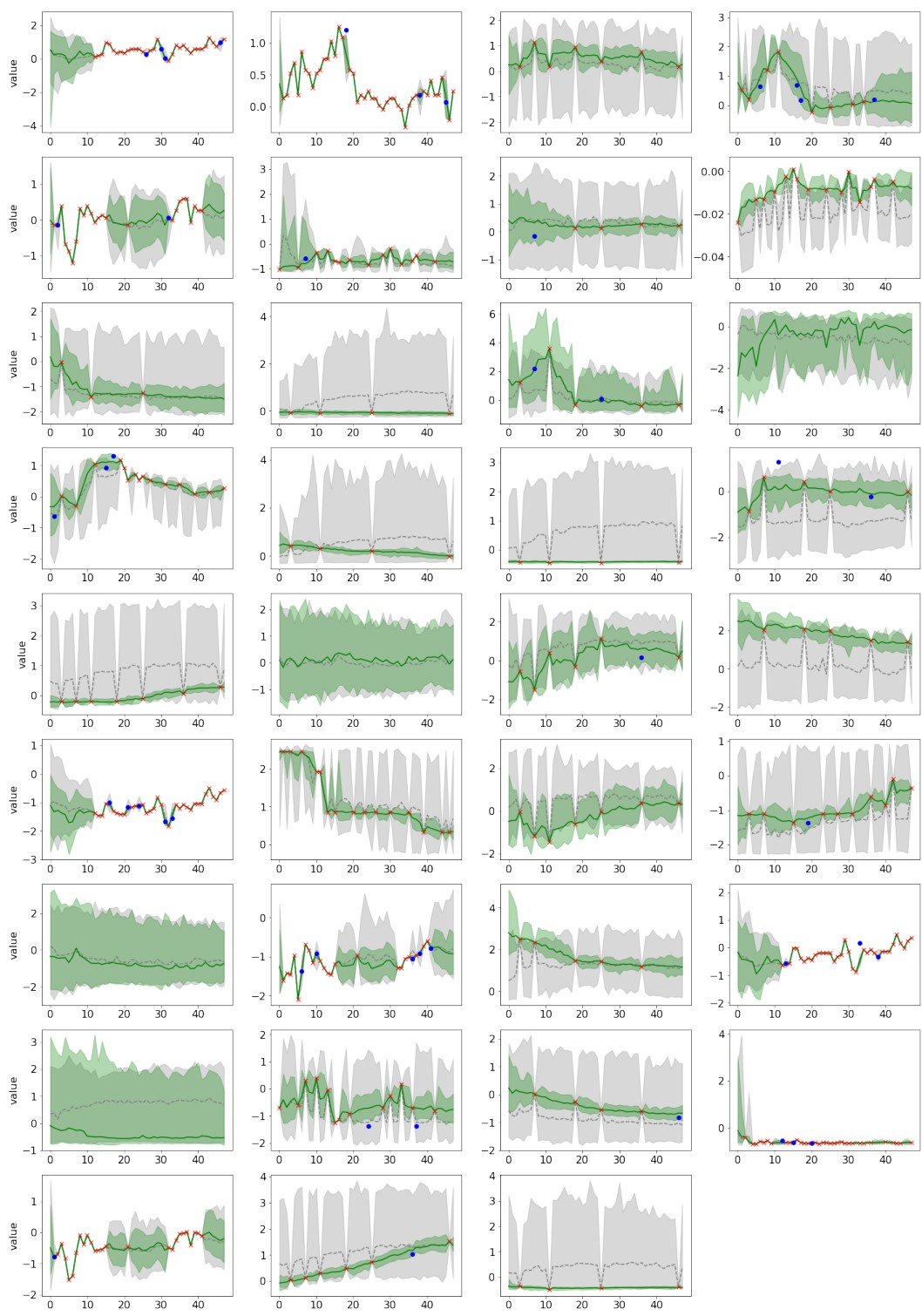

Figure 12: Comparison of imputation between the unconditional diffusion model and CSDI for the healthcare dataset (10% missing). The result is for a time series sample with all 35 features. The red crosses show observed values and the blue circles show ground-truth imputation targets. Green and gray colors correspond to CSDI and the unconditional model, respectively. For each method, median values of imputations are shown as the line and 5% and 95% quantiles are shown as the shade.

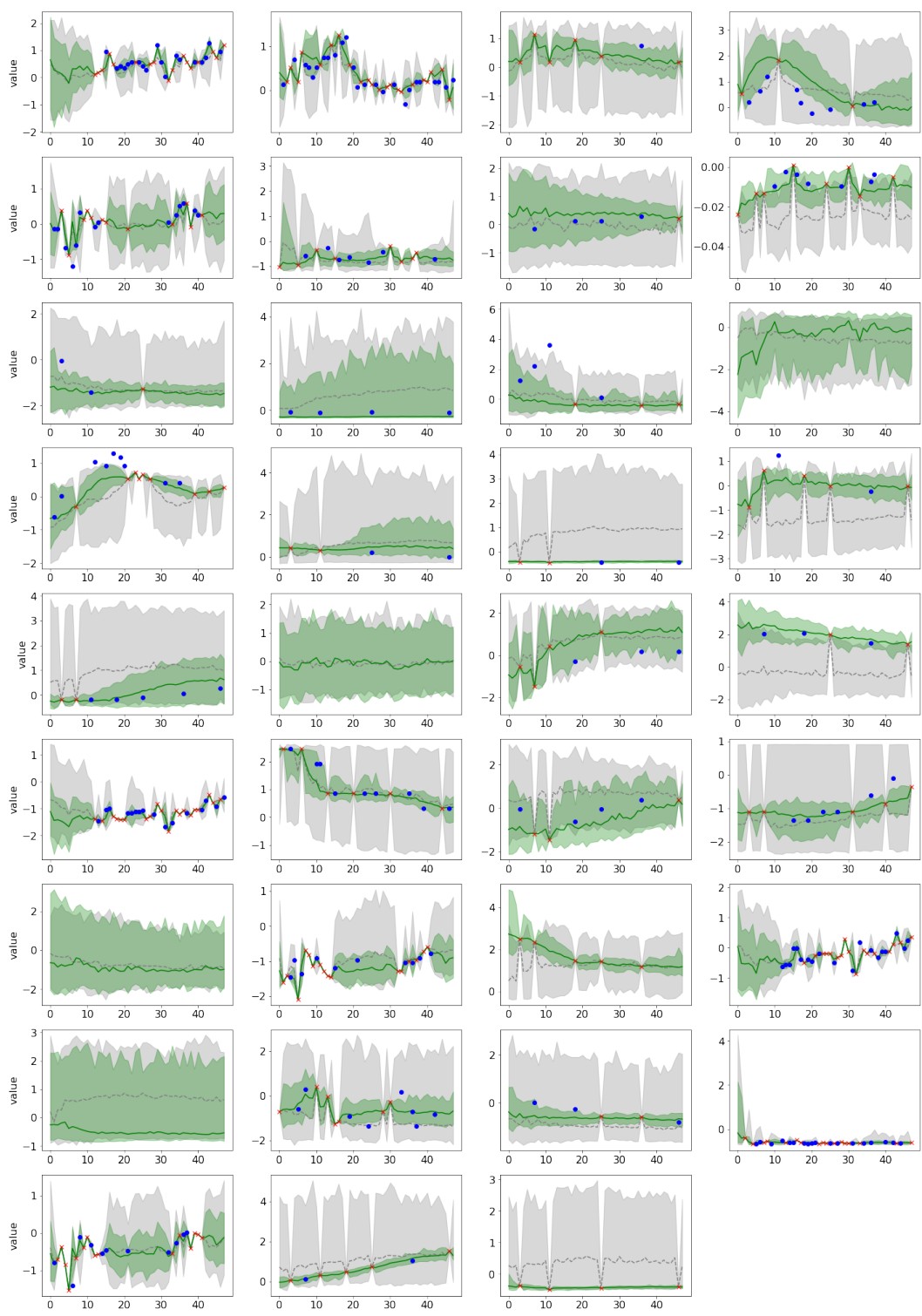

Figure 13: Comparison of imputation between the unconditional diffusion model and CSDI for the healthcare dataset (50% missing). The result is for a time series sample with all 35 features. The red crosses show observed values and the blue circles show ground-truth imputation targets. Green and gray colors correspond to CSDIand the unconditional model, respectively. For each method, median values of imputations are shown as the line and 5% and 95% quantiles are shown as the shade.

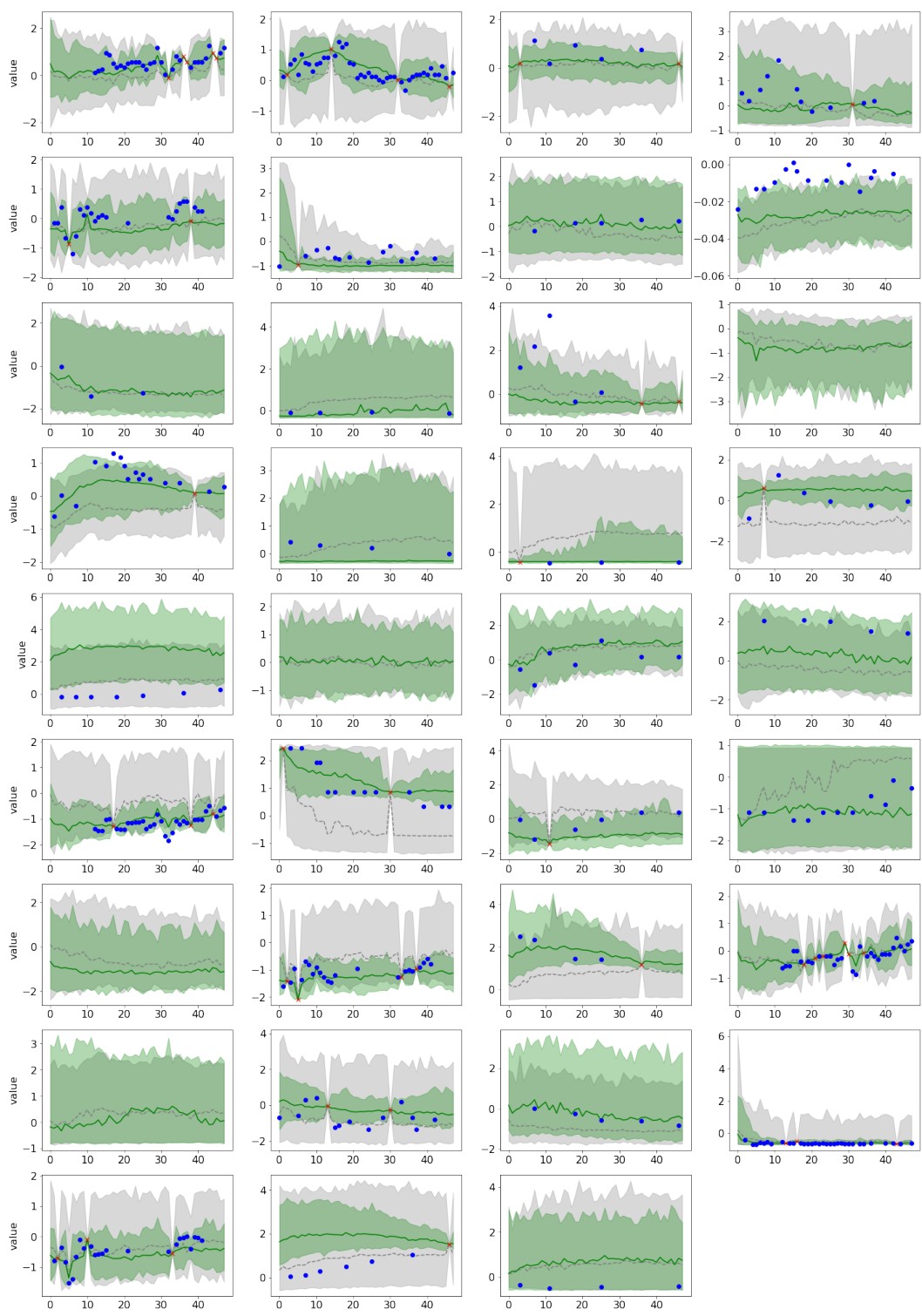

Figure 14: Comparison of imputation between the unconditional diffusion model and CSDI for the healthcare dataset (90% missing). The result is for a time series sample with all 35 features. The red crosses show observed values and the blue circles show ground-truth imputation targets. Green and gray colors correspond to CSDI and the unconditional model, respectively. For each method, median values of imputations are shown as the line and 5% and 95% quantiles are shown as the shade.

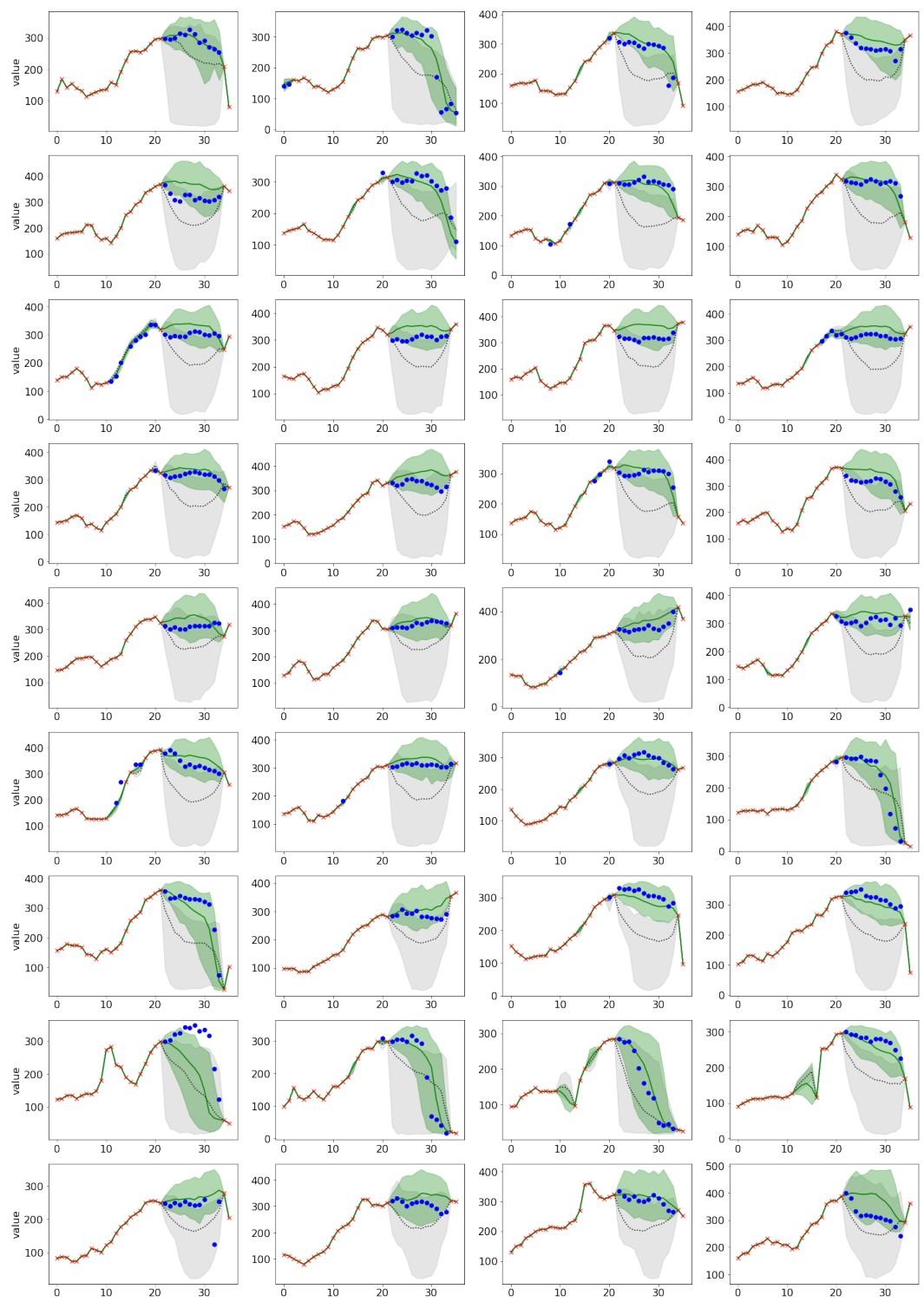

Figure 15: Comparison of imputation between the unconditional diffusion model and CSDI for the air quality dataset. The result is for a time series sample with all 36 features. The red crosses show observed values and the blue circles show ground-truth imputation targets. Green and gray colors correspond to CSDI and the unconditional model, respectively. For each method, median values of imputations are shown as the line and 5% and 95% quantiles are shown as the shade.