# OpenReview forum: "CSDI: Conditional Score-based Diffusion Models for Probabilistic Time Series Imputation"
_NeurIPS.cc/2021/Conference — NeurIPS 2021 Poster_

### Official Review · Reviewer_8rwP · 2021-07-16

**Rating:** 6
**Confidence:** 3

**Summary:**

The authors use diffusion probabilistic models for imputation / conditional generation and propose self-supervised training schemes for this task.

The authors demonstrate their method outperforms other state-of-art approaches for time-series data


**Ethical Concerns:**

The authors discuss societal / ethical impacts in Appendix H.

**Limitations And Societal Impact:**

The authors discuss societal / ethical impacts in Appendix H.

**Main Review:**

- Originality:
  - This paper explains a novel albeit incremental application of existing techniques.
  - Inpainting/ imputation/ infilling using score-matching has previously been explored for images and recently also for time-series, specifically for audio data [1]. Similarly [1] uses a mask for conditional generation, though this is not the focus of the paper.
  - Class conditioned generative modeling has been explored previously [2], where the score network is augmented by a class-dependent embedding. This paper’s approach may be seen as an extension of this but instead based on an embedding with inputs of observed/conditional data rather than class.
  - Modeling missing data as masked is standard for inpainting [1,3,4,5] and generating masks for self-supervised training has been performed before in other domains e.g. NLP. but not in combination with score-matching or time-series imputation as far as I am aware.

- Quality
  - The authors’ experimental section is excellent and they compare their approach to many state of the art methods in a variety of domains.

- Clarity
  - The paper is in general very well written.
  - A few important details are left for the appendices but are helpful for clarity and understanding the paper, specifically:
    - Definition of $\mu^{DDPM}$, line 97
    - The details on padding to ensure being same dimension inputs into the score network

- Significance
  - Although the individual techniques are not novel, the combination may be novel
  - The method is general, shows state of art performance and may be used in practice

Summary

Strengths:
 - Excellent experimental section and comparisons, the proposed method demonstrates state of art performance
 - Well written and clear paper

Weaknesses:
 - The work lacks significant novelty and is essentially an application of two existing methods 1) self-supervised training and 2) diffusion probabilistic models.  Score based models have previously been applied to time-series imputation and masked self-supervised training is fairly standard in training sequential models in NLP. I find it surprising that self-supervised techniques have not before been applied to train models for time-series imputation, as it would be an obvious extension of previous data imputation methods using pre-selected masks.

- Missing reference [1]

In summary, although there is nothing incorrect with the paper and the paper demonstrates good empirical results, unfortunately I believe the work is too incremental and lacks sufficient novelty to justify acceptance.

References
[1] Mittal, Gautam, et al. "Symbolic music generation with diffusion models." arXiv preprint arXiv:2103.16091 (2021). \
[2] Nichol, Alex, and Prafulla Dhariwal. "Improved denoising diffusion probabilistic models." arXiv preprint arXiv:2102.09672 (2021). \
[3]  Jinsung Yoon, James Jordon, Mihaela Schaar “GAIN: Missing Data Imputation using Generative Adversarial Nets” PMLR 80:5689-5698, 2018. \
[4] Luo, Yonghong, et al. "Multivariate time series imputation with generative adversarial networks." Proceedings of the 32nd International Conference on Neural Information Processing Systems. 2018. \
[5] Guo, Zijian, Yiming Wan, and Hao Ye. "A data imputation method for multivariate time series based on generative adversarial network." Neurocomputing 360 (2019): 185-197.

**Time Spent Reviewing:**

4

---

> ### Author Response · Authors · 2021-08-10
> **Response to Reviewer 8rwP**
>
> Thank you for the detailed review and thoughtful feedback. Below we address comments.
>
> __Q1.__ Imputation using score-matching has previously been explored, and a previous paper [1] uses a mask for conditional generation.
>
> __A1.__ Yes, in Sec. 3.3 and Appendix C.1, we discussed the existing imputation approach using score-matching. While [1] used a mask for conditional generation, the method in [1] utilized unconditional diffusion models and was equivalent with that discussed in Sec. 3.3 and Appendix C.1. In other words, masks in [1] were just used for selecting missing pixels at each sampling step. In contrast, CSDI utilizes conditional diffusion models, and masks in CSDI are used for self-supervised training. Namely, the purpose of using masks is quite different. In the results in Table 2 and 3 of the paper, we have shown this difference is crucial for imputation performance by comparing CSDI with unconditional diffusion models.
> Thank you for pointing out a missing reference. We will refer to [1] in Sec. 3.3 in the final paper.
>
> __Q2.__ This paper’s approach may be seen as an extension of Class conditioned generative modeling in [2] but instead based on an embedding with inputs of observed/conditional data rather than class.
>
> __A2.__ Yes, we can consider our approach as an extension of generative models conditioned on information whose shape is fixed for all samples, but our approach has a different purpose from the previous one. CSDI allows the shape of conditional information and inputs to be changeable per sample. In other words, whereas the previous approach only learns a conditional distribution, our approach can learn arbitrary conditional distributions simultaneously. By this extension, CSDI can significantly improve the imputation performance, as mentioned in the answer A1.
>
> We note that score-based generative modelings conditioned on fixed information are a popular approach. For example, DiffWave [3] utilized mel spectrogram as conditional information in a similar manner as [2].
>
>
> __Q3.__ The work lacks significant novelty and is essentially an application of two existing methods 1) self-supervised training and 2) diffusion probabilistic models.
>
> __A3.__ We would like to emphasize that the combination of self-supervised training and the conditional diffusion model is not a trivial task.
>
> 1. conditional diffusion models are novel because existing works focus on modeling an unconditional distribution, whereas we introduced CSDI with the attention mechanism to enable efficient conditional modeling. We described the importance of conditional diffusion models in the answer 1.
> 2. the mask that we employ is different from what is used in computer vision (CV) and NLP domains:
>     * In CV, generative models such as autoregressive models design their task with autoregressive structure, i.e. predict 1 step ahead.
>     * In NLP, language models like BERT [4] mask entire words / sentences (i.e. for a time t, the token in time t is either masked entirely or not masked at all).
>     * In multivariate time series, previous works (e.g. [5,6]) used pre-selected masks for representing missing indices. While a recent work [7] considered random masks for self-training, the random masks were fixed during training.
>
>     On the other hand, in ours, it is possible to mask anywhere per training sample and epoch. For example, for a time t, we can arbitrarily choose to mask part of it, which allows us to use partially observable features during the prediction. In particular, the mask strategy we provided can be utilized to specify which conditional distributions the model should learn (e.g. historical missing patterns for the air quality dataset and fixed missing patterns for forecasting tasks). This mask strategy enables us to train a suitable conditional diffusion model for a given dataset. In other words, the combination with the self-supervised training makes conditional diffusion models valuable.
>
> With the two contributions, our method is sufficiently novel, and empirical results in Section 6 suggest that.
>
> We would also like to stress that the proposed attention mechanism with temporal and feature transformer layers is another contribution of this work. In an ablation study in the rebuttal, we have compared the proposed attention mechanism with other variants and architectures and have shown that the proposed attention mechanism plays a key role for the large improvement of the imputation performance. For details of the ablation, please see the answer A1 in [this response](https://openreview.net/forum?id=VzuIzbRDrum&noteId=FzyhU4TZma).
>
> __Q4.__ Modeling missing data as masked is standard for inpainting and generating masks for self-supervised training has been performed before in other domains e.g. NLP.
>
> __A4.__ We agree that modeling missing data as masked is standard in previous studies in the time series domain, but the purpose of the mask is quite different from ours as described in the answer A3.
> In addition, we believe that the effectiveness of self-supervised training for multivariate time series imputation tasks is not obvious for the following reasons.
>
> 1. While self-supervised training with random masks is popular in the NLP domain, the training method has not been explored in the time series domain before, as you commented.
> 2. As discussed in the answer A3, the mask used in NLP is also different from ours. Moreover, we provided the mask strategy to learn suitable conditional distributions.
>
>
> __Q5.__ A few important details (the definition of $\mu_{DDPM}$ and the details on padding to ensure same dimension inputs) are left for the appendices but are helpful for clarity and understanding the paper.
>
> __A5.__ Thank you for the suggestion. For the second point, we will add the details on padding described in Section D to Section 5.
>
> For the first point, we agree that the details of DDPM including the definition of $\mu_{DDPM}$ could be helpful, but we consider the definition is not necessary since we do not focus on the form of $\mu_{DDPM}$ and we deal with $\mu_{DDPM}$ as a black-box function of the denoising function $\varepsilon_\theta$. Therefore, due to the page limitation, we chose to move the definition of $\mu_{DDPM}$ to the appendix. We will reconsider whether we can include this point in the final version of the main paper.
>
>
>
> [1] Mittal, Gautam, et al. "Symbolic music generation with diffusion models." arXiv preprint arXiv:2103.16091 (2021).
> [2] Nichol, Alex, and Prafulla Dhariwal. "Improved denoising diffusion probabilistic models." ICML, 2021.
> [3] Zhifeng Kong et al., "DiffWave: A Versatile Diffusion Model for Audio Synthesis", ICLR, 2021.
> [4] Jacob Devlin et al. BERT: Pre-training of deep bidirectional transformers for language understanding. In Proceedings of the 2019.   Conference of the North American Chapter of the Association for Computational Linguistics: Human Language Technologies, 2019.
> [5] Wei Cao et al. BRITS: Bidirectional recurrent imputation for time series. In Advances in Neural Information Processing Systems, 2018.
> [6] Jinsung Yoon et al. GAIN: Missing Data Imputation using Generative Adversarial Nets. PMLR 80:5689-5698, 2018.
> [7] Tae-Min Choi et al. RDIS: Random drop imputation with self-training for incomplete time series data. In The Thirty-Fifth AAAI Conference on Artificial Intelligence, 2021.

---

> > ### Comment · Reviewer_8rwP · 2021-08-23
> > **Response Response**
> >
> > Thank you for your response. After reconsideration I will raise my score.
> >
> > I tend to agree with reviewer yacf that the methodological contribution is slightly limited. However, the paper offers a few novel contributions (attention architecture for conditioning, random mask strategy) and given the empirical performance shown, I believe the method tackles an important issue and can be used in an applied setting.

---

### Official Review · Reviewer_2d7W · 2021-07-17

**Rating:** 6
**Confidence:** 4

**Summary:**

This paper focuses on the problem of the probabilistic imputation of missing data in time series. This work proposes a novel approach for probabilistic imputation based on the score-based diffusion models. Experiments on time series imputation tasks show that the proposed method achieves better performance than the current SOTA methods in this space.

**Limitations And Societal Impact:**

Yes.

**Main Review:**

### Novelty and Significance
- The problem addressed is very important in many domains. The proposed approach is novel.

### Quality
- The paper is missing comparisons with some standard baselines for probabilistic imputation/interpolation such as Single-Task [1] and Multi-task Gaussian Processes [2] and SOTA methods based on neural ODEs [3]. ODE methods also perform particularly well on forecasting tasks so should be compared in Table 5 as well. Furthermore, the authors are also missing a comparison with mTANs in Tables 2 and 3 considering mTANs can produce probabilistic interpolation by sampling multiple latent states and passing through the decoder.
- Negative log-likelihood is a standard metric for evaluation uncertainty computations. What is the reason behind omitting such a standard metric when evaluating probabilistic interpolations and imputations?
- The authors should perform ablation to show the advantage of using temporal and feature transformer layers.

### Clarity
- The paper is well written and easy to follow.

### Additional Comments
- How are the interpolation and imputation tasks on the PhysioNet dataset different after performing discretization on the irregularly sampled dataset? Is the proposed approach applicable for handling irregularly sampled sequences directly since the discretization loses a lot of information about the time series?

#### References
1. C. E. Rasmussen and C. K. I. Williams. Gaussian processes for machine learning. Adaptive computation and machine learning. 2006.
2. E. V. Bonilla, K. M. Chai, and C. Williams. Multi-task Gaussian process prediction. In Advances in Neural Information Processing Systems, pages 153–160, 2008.
3. Y. Rubanova, R. T. Q. Chen, and D. K. Duvenaud. Latent ordinary differential equations for irregularly sampled time series. In Advances in Neural Information Processing Systems, pages 5320–5330. 2019.

**Time Spent Reviewing:**

5

---

> ### Author Response · Authors · 2021-08-10
> **Response to Reviewer 2d7W**
>
> ​​Thank you for the detailed review and thoughtful feedback. Below we address comments and questions. We will add ablations in the final paper. Due to the time limitation, we have fixed the missing ratio to 10% in the healthcare dataset for most of the ablations and run each experiment three times.
>
> __Q1.__ An ablation to show the advantage of using temporal and feature transformer layers should be performed.
>
> __A1.__ We appreciate your suggestion. As an ablation, we have evaluated the advantage of our attention mechanism that uses temporal and feature transformer layers. Concretely, we compared our attention mechanism with the following variants and architectures for imputation tasks.
>
> * no temporal: remove temporal attention layers
> * no feature: remove feature attention layers
> * flatten: flatten 2D tensor ($K$ features x $L$ length) to 1D, and input the 1D vector to transformer layers
> * Bi-RNN: replace the attention with Bi-directional RNN which is a popular architecture for multivariate time series imputation.
> * dilated conv: replace temporal and feature attention layers with 1D dilated convolution layers, respectively. The dilated convolution was used in previous studies for diffusion models [1,2].
>
> We set hyperparameters so that the number of parameters becomes similar to the proposed attention mechanism. The result is shown in the table below.
>
> |  | MAE(health) | CRPS(health) | MAE(air) | CRPS(air) |
> | :--- | ---: | ---: | ---: | ---: |
> | no-cross | 0.352 | 0.386 | 14.44 | 0.162 |
> | no-temporal | 0.439 | 0.475 | 26.63 | 0.292 |
> | flatten | 0.383 | 0.418 | 12.26 | 0.139 |
> | Bi-RNN | 0.272 | 0.301 | 12.56 | 0.142 |
> | dilated conv | 0.279 | 0.305 | 11.67 | 0.130 |
> | proposed | __0.217__ | __0.238__ | __9.60__ | __0.108__ |
>
> (* health:healthcare dataset (missing ratio:10%), air: air quality dataset)
>
> Our proposed attention mechanism outperforms all of the other architectures. The comparison with “no temporal” and “no feature” shows both temporal and feature correlations are important for accurate imputation. The comparison with “flatten”, “Bi-RNN”, and “dilated conv” shows our attention mechanism is effective to learn temporal and feature dependency compared with existing methods. These results indicate the proposed attention mechanism plays a key role in improving the imputation performance by a large margin.
>
> __Q2.__ The comparison with some standard baselines for probabilistic forecasting is missed.
>
> __A2.__ Thank you for the indication. As an ablation, we chose the multi-task Gaussian process (MT-GP) as a standard baseline and compared CRPS between MT-GP and the proposed CSDI for probabilistic imputation. The following result shows that CSDI outperforms MT-GP by a large margin.
>
> |  | healthcare (miss:10%) | Air Quality |
> | :---: | ---: | ---: |
> | MT-GP | 0.483 | 0.304 |
> | CSDI | __0.238__ | __0.108__ |
>
>
> __Q3.__ The comparison with interpolation methods based on Neural ODEs is missed.
>
> __A3.__ A reason why we did not include these methods in the paper is that a previous study [3] already showed mTANs outperformed Latent ODE [4]. To confirm the advantage of CSDI, we performed an ablation comparing CSDI with Latent ODE for interpolation and forecasting tasks. For forecasting tasks, we chose the solar dataset. Since Latent ODE did not converge when we used all features, we only used 20 features.
>
> We show the result as follows. CSDI outperforms latent ODE for both interpolation and forecasting tasks.
>
> |  | CRPS(health) | MAE(health) | CRPS-sum(solar) | MAE(solar) |
> | :---: | ---: | ---: | ---: | ---: |
> | Latent ODE | 0.704 | 0.524 | 0.383 | 26.31 |
> | CSDI | __0.380__ | __0.362__ | __0.278__ | __14.69__ |
>
> (* health: interpolation for the healthcare data (missing:10%), solar: forecasting for the solar data (20 features))
>
>
> __Q4.__ The comparison with mTANs as probabilistic methods is missed.
>
> __A4.__  We appreciate your pointing out. As an ablation, we compared CSDI with mTANs in terms of probabilistic interpolation. We ran mTANs 100 times and calculated CRPS. The following table demonstrates the comparison of CRPS for the healthcare dataset. CSDI achieves lower CRPS than mTANs. The differences between two methods are larger than the differences in MAE in Table 4 of the paper. This means that CSDI outperforms mTANs rather as probabilistic interpolation methods. We observed that the estimated distribution by mTANs is too tight compared to the data distribution.
>
> |  | missing:10% | missing:50% | missing:90% |
> | :---: | ---: | ---: | ---: |
> | mTANs | 0.526 | 0.567 | 0.689 |
> | CSDI | __0.380__ | __0.418__ | __0.556__ |
>
> We note that we also re-calculated MAE as the median of 100 samples and confirmed that the change in MAE from Table 4 is marginal.
>
> __Q5.__ What is the reason behind omitting negative log likelihood (NLL) when evaluating probabilistic interpolations and imputations?
>
> __A5.__  The reason why we did not adopt NLL is that ELBO we minimized is sometimes far from NLL and uncorrelated with the quality of generated samples. Specifically, in the proposed method, the choice of the noise schedule highly affects the ELBO while it has little effect on the sample quality.
>
> To address this, we performed an ablation study. We chose the following three noise schedules for CSDI and calculated NLL and CRPS for each schedule.
>
> * quadratic (used in the paper): quadratic spaced schedule between $\beta_\min=0.0001$ and $\beta_\max=0.5$
> * linear: linear spaced schedule with the same $\beta_\min$ and $\beta_\max$ as those in the paper
> * quadratic (large minimum noise): quadratic schedule with large minimum noise level $\beta_\min=0.001$, which makes the model ignore small noise
>
> We also calculated NLL for GP-VAE.
> The result is shown in the following table. While CRPS by the proposed method is almost independent from the choice of schedules, NLL significantly depends on the schedule. This phenomenon happens because time series data is generally noisy and it is difficult to denoise small noise during imputation. Estimated scores by the model could be inaccurate when inputs to the model (i.e. imputation targets) only contain small noise. These inaccurate scores could make the estimated ELBO loose, whereas small noise does not affect the sample quality. When the minimum noise level $\beta_\min$ is large, since the model does not denoise small noise in sampling steps, ELBO by the proposed method is tightly estimated and smaller than that by GP-VAE.
>
> Therefore, ELBO (and NLL) is not suitable for evaluating the sample quality and we adopted other metrics such as CRPS and MAE.
>
> | method | noise schedule | NLL(health) | CRPS(health) | NLL(air) | CRPS(air) |
> | :--- | :--- | ---: | ---: | ---: | ---: |
> | GP-VAE | - | < 1.22 | 0.574 | < 1.09 | 0.397 |
> | proposed | quadratic (used in paper) | < 1.63 | __0.238__ | < 0.97 | __0.108__ |
> | proposed| linear | < 29.70 | 0.240 | < 18.55 | 0.110 |
> | proposed| quadratic (large minimum noise) | __< 0.07__ | 0.239 | __< -0.70__ | 0.109 |
>
> (* health: healthcare dataset (missing ratio:10%), air: air quality dataset)
>
> We note that how to set a better noise schedule is an ongoing research topic [5], and we leave how to set a schedule to ensure both the sample quality and tighter ELBO in future work.
>
> __Q6.__ How are the interpolation and imputation tasks on the PhysioNet dataset different after performing discretization?
>
> __A6.__ Methodologically, there is no difference between two tasks after the discretization. The only difference is the length of each sample of time series. The maximum length is 48 on the imputation task, while 202 on the interpolation task. The difference of length affects the computational cost and memory usage. In particular, since we adopt the attention mechanism, the computational cost quadratically increases depending on the length of time series.
>
> __Q7.__ Is the proposed approach applicable for handling irregularly sampled sequences directly?
>
> __A7.__ As with the answer A6, methodologically yes. The proposed approach can handle raw irregularly sampled sequences thanks to the attention mechanism with time embedding. In practice, when each sample of time series comprises many observation points (e.g., several thousands), the computational cost and memory usage could be an issue. In this case, a possible solution is the use of efficient attention mechanisms (e.g. [6]) which have recently been developed to improve computational efficiency.
>
> [1] Zhifeng Kong et al., "DiffWave: A Versatile Diffusion Model for Audio Synthesis", ICLR, 2021.
> [2] Kashif Rasul et al., "Autoregressive denoising diffusion models for multivariate probabilistic time series forecasting", ICML, 2021.
> [3] Satya Narayan Shukla and Benjamin M Marlin. Multi-Time Attention Networks for Irregularly Sampled Time Series, ICLR, 2021.
> [4] Y. Rubanova et al. Latent ordinary differential equations for irregularly sampled time series. NeurIPS, 2019.
> [5] Alex Nichol and Prafulla Dhariwal. Improved Denoising Diffusion Probabilistic Models, ICML, 2021.
> [6] Zhuoran Shen et al. Efficient attention: Attention with linear complexities. In Proceedings of the IEEE/CVF Winter Conference on Applications of Computer Vision, 2021.

---

> > ### Comment · Reviewer_2d7W · 2021-08-23
> > **Thank you.**
> >
> > Thank you for addressing my comments, make sure to include them in the paper. Although there are some novelty issues raised by other reviewers, I am inclined towards accepting this paper as it addresses an important problem and improves over the previous state-of-the-art methods.

---

### Official Review · Reviewer_yacf · 2021-07-17

**Rating:** 6
**Confidence:** 4

**Summary:**

This work proposes a probabilistic data imputation model for multivariate time series. The proposed method uses a conditional diffusion model to estimate the (masked) conditional distribution of missing data given observed values. To utilise information from observed values, the authors employ a neural attention mechanism to capture spatiotemporal dependencies of time series. The authors use self-supervised learning for training the model.  In experiments, the author demonstrate competitive performance of their method against a number of comparable methodologies on time series imputation and forecasting tasks.


**Limitations And Societal Impact:**

My main concern with the work is that methodologically, its contribution is rather limited. The main methodological contribution of this work is the modification of denoising function parameterising mu in eq (5), which in my view is a straightforward and minimal extension of DDPM. The architectural modification involving the use of neural attention layer appears to be interesting, but without any comparison it is difficult to say how crucial it is to the performance of the model as opposed to for instance using (bidrectional) RNNs.


**Main Review:**

The proposed model is an extension of the previously proposed denoising diffusion probabilistic model (DDPM), where the Markov chain reverse process of DDPM is conditioned on observed data. More specifically, the denoising function that parameterises the mean of the conditional Markov chain distributions in the DDPM reverse process is provided with the observed data as an additional argument.

The paper is well written and easy to follow. The proposed method builds on a methodology that has been shown to be competitive in a variety of domains including time series modelling. The task addressed by the authors is quite relevant. The utilisation of attention mechanism for capturing both spatial and temporal dependencies appears interesintg. In comparison to recent relevant methodologies, the method has been shown to be state-of-the-art on time series imputation and forecasting tasks involving a number of benchmark datasets.


**Time Spent Reviewing:**

6

---

> ### Author Response · Authors · 2021-08-10
> **Response to Reviewer yacf**
>
> Thank you for the detailed review and thoughtful feedback. Below we address comments.
>
> __Q1.__ Methodologically, the contribution is limited.
>
> __A1.__  We would like to emphasize that the combination of self-supervised training and the conditional diffusion model is also a key contribution.
>
> The mask that we employ for self-supervised training is different from what is used in other domains such as computer vision (CV) and NLP:
>
> * In CV, generative models such as autoregressive models design their task with autoregressive structure, i.e. predict 1 step ahead.
> * In NLP, language models like BERT mask entire words / sentences (i.e. for a time t, the token in time t is either masked entirely or not masked at all).
>
> On the other hand, in ours, it is possible to mask anywhere. For example, for a time t, we can arbitrarily choose to mask part of it, which allows us to use partially observable features during the prediction. In particular, the mask strategy we discussed can be utilized to specify which conditional distributions the model should learn (e.g. historical missing patterns for the air quality dataset and fixed missing patterns for forecasting tasks). This mask strategy enables us to train a suitable conditional diffusion model for a given dataset. In other words, the combination with the self-supervised training makes conditional diffusion models valuable.
>
> __Q2.__ It is difficult to say how crucial the proposed attention mechanism is to the performance without any comparison.
>
> __A2.__ Thank you for pointing it out. We conducted an ablation study to show how crucial our attention mechanism with temporal and feature transformer layers is. We compared our attention mechanism with the following variants and architectures for imputation tasks:
>
> * no temporal: remove temporal attention layers
> * no feature: remove feature attention layers
> * flatten: flatten 2D tensor ($K$ features x $L$ length) to 1D, and input the 1D vector to transformer layers
> * Bi-RNN: replace the attention mechanism with Bi-directional RNN which is a popular architecture for multivariate time series imputation.
> * dilated conv: replace temporal and feature attention layers with 1D dilated convolution layers, respectively. The dilated convolution was used in previous studies for diffusion models [1,2].
>
> We set hyperparameters so that the number of parameters becomes similar to the proposed attention mechanism. The result is shown in the table below.
>
> |  | MAE(health) | CRPS(health) | MAE(air) | CRPS(air) |
> | :--- | ---: | ---: | ---: | ---: |
> | no-cross | 0.352 | 0.386 | 14.44 | 0.162 |
> | no-temporal | 0.439 | 0.475 | 26.63 | 0.292 |
> | flatten | 0.383 | 0.418 | 12.84 | 0.147 |
> | Bi-RNN | 0.272 | 0.301 | 12.56 | 0.142 |
> | dilated conv | 0.279 | 0.305 | 11.67 | 0.130 |
> | proposed | __0.217__ | __0.238__ | __9.60__ | __0.108__ |
>
> (* health:healthcare dataset (missing ratio:10%), air: air quality dataset. Each ablation was held three times)
>
> Our proposed attention mechanism outperforms all of the other architectures. The comparison with “no temporal” and “no feature” shows that both temporal and feature correlations are important for accurate imputation. The comparison with “flatten”, “Bi-RNN”, and “dilated conv” shows that our attention mechanism is effective to learn temporal and feature dependency compared with existing methods.
> In summary, the result of the ablation indicates the proposed attention mechanism plays a key role in improving the imputation performance by a large margin.
>
> [1] Zhifeng Kong et al., "DiffWave: A Versatile Diffusion Model for Audio Synthesis", ICLR, 2021.
> [2] Kashif Rasul et al., "Autoregressive denoising diffusion models for multivariate probabilistic time series forecasting", ICML, 2021.

---

### Author Response · Authors · 2021-08-10
**A summary of responses**

We would like to thank all reviewers for providing high quality reviews and insightful feedback. We are encouraged that reviewers think the problem we address is “very important in many domains” (R2); our proposed method “is novel” (R2), “is general” (R3), “shows state-of-the-art performance” (R1,R3), and “may be used in practice” (R3); our utilization ​​of attention mechanism for capturing both spatial and temporal dependencies “appears interesting” (R1); our experimental section is “excellent” (R3); and our paper is “well-written” (R1,R2,R3), “clear” (R3), and “easy to follow” (R1,R2).

(We abbreviate the reviewer yacf, 2d7W, 8rwP to R1, R2, R3, respectively.)

In response to comments from reviewers, we conducted ablation studies. The list of ablations is described below. We will add these ablations in the final paper.

* Comparison between the proposed attention mechanism and various other architectures to show how crucial the proposed attention mechanism is  (in response to R1 and R2)
* Comparison with interpolation/forecasting methods based on Neural ODEs (to R2)
* Comparison with a basic baseline (Multi-task GP) for probabilistic imputation (to R2)
* Comparison with mTANs as probabilistic interpolation tasks (to R2)
* Comparison of negative log likelihood for probabilistic imputation (to R2)

Note that we had a typo in Table 2 in the paper. The result of CRPS obtained by CSDI for the air quality dataset should be 0.108 instead of 0.113. It doesn't affect our claims.

---

### Decision · Program_Chairs · 2021-09-27

**Decision:**

Accept (Poster)

**Comment:**

This paper uses conditional score-based diffusion models for probabilistic time series imputation. The authors use an attention mechanism to capture the temporal and feature dependencies. A self-supervised training method is developed. The paper is overall well-written and the method appears to outperform alternative imputation techniques on some metrics.

The proposed method is novel but was also perceived as a rather straightforward combination of existing techniques (conditional score ideas have already been used in different domains and the self-supervised technique has been applied in related areas). That said, it is by no means trivial to put this combination in practice and the empirics are compelling.

After the rebuttal of the authors, some of the reviewers increase their score. With scores of 6, 6, 6, there remains a lack of enthusiasm for the paper. However, I recommend acceptance of the paper as it is a useful contribution to probabilistic time series imputation.